# Single-cell glycomics analysis by CyTOF-Lec reveals glycan features defining cells differentially susceptible to HIV

**Tongcui Ma[1,2], Matthew McGregor[1,2], Leila Giron[3], Guorui Xie[1,2], Ashley F George[1,2], Mohamed Abdel-Mohsen[3]\*, Nadia R Roan[1,2]\***

[1]Department of Urology, University of California, San Francisco, San Francisco, United States; [2]Gladstone Institutes, San Francisco, United States; [3]The Wistar Institute, Philadelphia, United States

**\*For correspondence:**
mmohsen@Wistar.org (MA-M);
nadia.roan@gladstone.ucsf.edu
(NRR)

**Competing interest:** The authors declare that no competing interests exist.

## Abstract
High-parameter single-cell phenotyping has enabled in-depth classification and interrogation of immune cells, but to date has not allowed for glycan characterization. Here, we develop CyTOF-Lec as an approach to simultaneously characterize many protein and glycan features of human immune cells at the single-cell level. We implemented CyTOF-Lec to compare glycan features between different immune subsets from blood and multiple tissue compartments, and to characterize HIV-infected cell cultures. Using bioinformatics approaches to distinguish preferential infection of cellular subsets from viral-induced remodeling, we demonstrate that HIV upregulates the levels of cell-surface fucose and sialic acid in a cell-intrinsic manner, and that memory CD4+ T cells co-expressing high levels of fucose and sialic acid are highly susceptible to HIV infection. Sialic acid levels were found to distinguish memory CD4+ T cell subsets expressing different amounts of viral entry receptors, pro-survival factors, homing receptors, and activation markers, and to play a direct role in memory CD4+ T cells' susceptibility to HIV infection. The ability of sialic acid to distinguish memory CD4+ T cells with different susceptibilities to HIV infection was experimentally validated through sorting experiments. Together, these results suggest that HIV remodels not only cellular proteins but also glycans, and that glycan expression can differentiate memory CD4+ T cells with vastly different susceptibility to HIV infection.

## Editor's evaluation
This study applies a new novel method of single cell detection to biologically relevant systems to try to understand whether glycans on the surface of CD4+ T cells impact HIV susceptibility. They find that cells expressing higher levels of fucose and sialic acid are more likely to be infected with HIV than those with low levels. The findings point to glycans as biomarkers and potential determinants for cellular susceptibility to HIV, and open the door to new avenues for studying the interplay between cell surface glycans and viral infections.

## Introduction
Viruses generally need to hijack multiple host cell processes to complete their replication cycle. This virus-mediated manipulation of host processes is called viral-induced remodeling, and has been studied using a variety of approaches, most entailing virally infected cell lines analyzed in bulk. More recently, viral remodeling of primary cells was studied at the single-cell level by characterizing, via cytometry by time of flight (CyTOF), human tonsillar T cells infected ex vivo by varicella zoster virus (VZV) (*Sen et al., 2014*). Bioinformatics analyses of high-dimensional CyTOF datasets of VZV-infected

**eLife digest** Living cells have a sugar coating. These sugars include molecules called glycans, which help cells interact with the outside world. The types of sugars on cells can affect their properties, including potentially their susceptibility to infection by viruses, such as the human immunodeficiency virus, HIV.

To date, most research examining cells susceptible to HIV has focused on cell surface proteins, not sugars. To study these proteins, researchers had previously covered them in metal-studded antibodies (which stick to proteins) and used a technique called cytometry time of flight, or CyTOF for short, to quantify the levels of these proteins on the surface of cells susceptible to HIV. Adapting this tool to investigate sugars could answer questions about HIV infection. For example, does the virus prefer to infect cells coated in certain sugar molecules? And does it change the pattern of sugars on the surface of the cells it infects?

Ma et al. adapted CyTOF to use molecules called lectins (which stick to sugars) in conjunction with the metal-studded antibodies. This made it possible to simultaneously measure the levels of 34 different proteins and 5 different types of sugars on individual cells. The pattern of sugars on the surface of cells from the immune system differed depending on what tissues the cells came from, and what types of cells they were. The results showed that HIV preferred to infect memory CD4 T cells with high levels of two types of sugar: fucose and sialic acid. Furthermore, during infection, the levels of both these sugars increased.

Current treatments for HIV keep virus levels low but do not cure the infection. Further research could determine whether sugars have a role to play in HIV persistence. It is possible that the sugar patterns preferred by the virus help it to avoid detection. A clearer understanding of cell surface sugars could lead to sugar-targeting drugs that kill infected cells.

---

cells revealed that VZV infection elicits significant host cell remodeling and alters the skin-trafficking property of subsets of infected cells. We recently implemented a follow-up approach, termed predicted precursor as determined by single-cell linkage using distance estimation (PP-SLIDE), to document HIV-induced remodeling of T cells from blood, lymph node, and genital tract, define the subsets of cells most susceptible to HIV infection, and characterize the phenotypes of HIV-infected cells in viremic and virally suppressed PLWH (*Cavrois et al., 2017*; *Ma et al., 2020*; *Neidleman et al., 2020b*; *Xie et al., 2021*).

One important feature of PP-SLIDE is that it enables assessment of whether a receptor differentially expressed on HIV-infected cells reflects HIV-induced remodeling or preferential infection of cells harboring that pattern of expression of the receptor. For example, PP-SLIDE established that HIV-infected T cells express low levels of CD4 and CD28 not because HIV preferentially infects CD4[Low]-CD28[Low] T cells, but rather because HIV downregulates these receptors (*Cavrois et al., 2017*; *Ma et al., 2020*; *Neidleman et al., 2020b*; *Xie et al., 2021*), which were independently shown to be down-modulated by HIV accessory genes (*Garcia and Miller, 1991*; *Swigut et al., 2001*). Other HIV-remodeled surface receptors identified by PP-SLIDE include those involved in T cell migration to lymph nodes and markers of Tfh cells (*Cavrois et al., 2017*; *Ma et al., 2020*; *Neidleman et al., 2020b*; *Xie et al., 2021*). By contrast, the low levels of surface CD127 expression on HIV-infected tonsillar T cells reflected preferential sparing of CD127[High] T cells from productive infection (*Cavrois et al., 2017*; *Ma et al., 2020*; *Neidleman et al., 2020b*; *Xie et al., 2021*). Subsequent studies demonstrated that CD127[High] memory T cells preferentially undergo latent infection by HIV (*Hsiao et al., 2020*). These and other PP-SLIDE-generated findings of preferential infection of cellular subsets have been experimentally validated through a variety of sorting experiments (*Cavrois et al., 2017*; *Ma et al., 2020*; *Neidleman et al., 2020b*; *Xie et al., 2021*). Together, these studies suggest that important insights into HIV pathogenesis and persistence can be gained from characterizing HIV-induced remodeling of primary cells at a single-cell level.

However, such remodeling studies – and in fact all phenotypic characterizations of virally infected cells to date – have only examined the cells' proteomes. Completely overlooked has been the diverse collection of glycans that are assembled on the surface of all living cells (*Williams and Thorson, 2009*). Cell-surface glycosylation plays critical roles in regulating multiple cellular processes and immune

functions (*Barrera et al., 2002*), as well as cell-cell (*de Freitas Junior et al., 2011*) and cell-pathogen (*Colomb et al., 2019*; *Everest-Dass et al., 2012*; *Giron et al., 2020b*) interactions. Furthermore, multiple viruses (e.g., HSV-1, CMV, and HTLV1) have been shown to alter the surface glycosylation of infected cells (*Hiraiwa et al., 2003*; *Kambara et al., 2002*; *Nyström et al., 2007*; *Nyström et al., 2009*). To date, studies of host glycomes have been limited to analysis of bulk cells, using techniques such as mass spectrometry, liquid chromatography, and lectin microarrays (*Chen et al., 2021*), although bacteria have been characterized at the single-cell level using lectins (*Leipold et al., 2011*). A recent study analyzing bulk populations of CD4+ T cells with different glycan features demonstrated that they harbor different levels of HIV transcripts (*Colomb et al., 2020*), suggesting that the host cell glycome can affect HIV susceptibility and/or replication. However, robust tools to deeply characterize, at the single-cell level, the glycan features of immune cells – including HIV-infected ones – are lacking to date.

In this study, we developed a new approach taking advantage of the high-parameter analysis capabilities of CyTOF (*Bendall et al., 2011*), to phenotype cells simultaneously for protein and glycan features. This was achieved through conjugating a collection of lectins (proteins that specifically bind different types of glycans) to metal lanthanides, an approach that has previously been validated at the bulk level (*Leipold et al., 2009*). We call our approach CyTOF-Lec – as it combines traditional CyTOF (using lanthanide-conjugated antibodies) with lanthanide-conjugated lectins to characterize surface glycosylation patterns of cells – and applied it on both blood and tissue cells. Taking advantage of the high-dimensional nature of our resulting CyTOF-Lec datasets and our PP-SLIDE analysis pipeline, we set out to address the following two fundamental questions about HIV infection: (1) Does HIV preferentially infect cells exhibiting distinct glycan features, and (2) to what extent does HIV remodel the glycan features of its host cell?

## Results
### Development and validation of CyTOF-Lec

To establish a methodology that could simultaneously characterize protein and glycan features at the single-cell level, we developed a panel of lanthanide metal-conjugated antibodies and lectins compatible with CyTOF, which we refer to as CyTOF-Lec (*Supplementary file 1* A). As tonsils provide an abundant source of both T and B cells, we used these cells for our initial validation of the panel. First, we confirmed that the staining patterns of the lanthanide-conjugated antibodies were consistent with the known differential expression of their target antigens on tonsillar T vs. B cells (*Figure 1—figure supplement 1*), and with results previously reported using CyTOF without lectin staining (*Ma et al., 2020*). To ensure that the lectin staining did not displace or alter the antibodies bound to their protein targets, we developed a protocol whereby surface antibody staining was completed prior to lectin staining (see Materials and methods). We confirmed that all five lectins (AOL, MAL-1, WGA, UEA-1, ABA, see *Supplementary file 1*) conjugated to lanthanides stained both T and B cells (*Figure 1—figure supplement 1*). Furthermore, we confirmed that antibody binding to CD3, CD4, and CD8 was the same whether or not the specimens were subsequently stained with lectins (*Figure 1A*). To establish the specificity of lectin binding, we assessed the effect of sialidase, which degrades cell-surface sialic acid. As expected, binding by WGA and MAL-1, which detect different forms of sialic acid (*Supplementary file 1*), was decreased after sialidase treatment of the cells (*Figure 1B*). By contrast, binding by AOL and UEA-1, which detect different forms of fucose, and ABA, which detects T antigen (*Supplementary file 1*), were all increased (*Figure 1C–D*). This was expected as removal of sialic acid should enable better detection of these other glycan structures, and is consistent with prior reports (*Giron et al., 2020b*).

To determine whether the conjugated lectins could detect differences in cell-surface glycans between subsets of immune cells, we compared binding by each of the five lectins to B cells, memory CD8+ T cells (CD8+ Tm cells), naïve CD8+ T cells (CD8+ Tn cells), memory CD4+ T cells (CD4+ Tm cells), and naïve CD4+ T cells (CD4+ Tn cells). We assessed the expression of glycans on these subsets not only among tonsillar cells, but also among PBMCs and endometrial T cells for comparison. For all three sites, the fucose-specific lectins AOL and UEA1 bound more to CD4+ Tm and CD8+ Tm cells than to their respective naïve counterparts, although the difference only reached significance for AOL binding in tonsils (*Figure 2A*). These results are consistent with

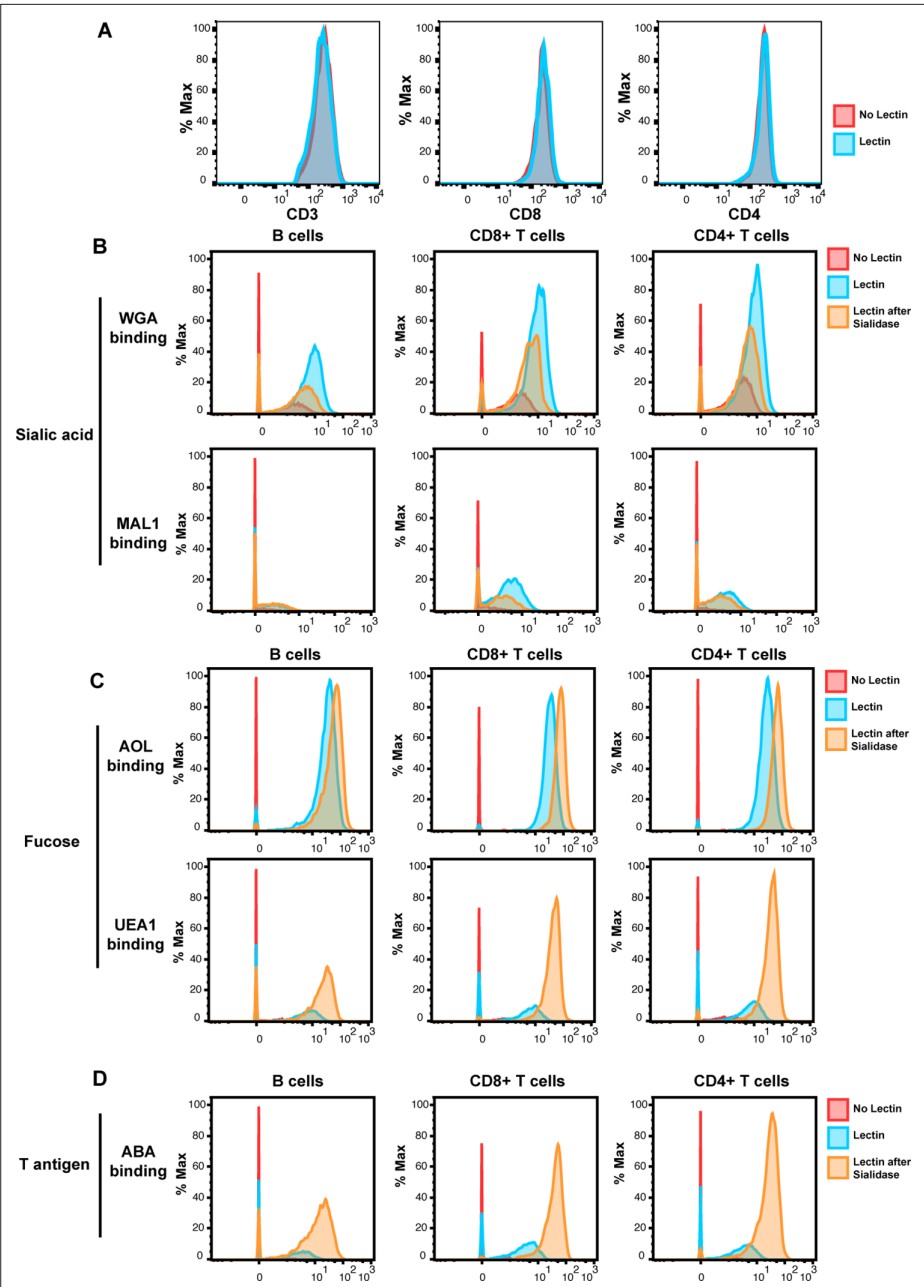

**Figure 1.** Validation of cytometry by time of flight (CyTOF)-Lec. (**A**) Antibody staining for protein markers is not altered by lectins. Shown are histograms of tonsil cells expressing CD3, CD8, or CD4, as detected by CyTOF after antibody staining followed or not by staining with metal-conjugated lectins (AOL: *Aspergillus oryzae*; MAL-1: *Maackia amurensis* I; WGA: wheat germ agglutinin; UEA-1: *Ulex europaeus* I; and ABA: *Agaricus bisporus* agglutinin). Protein expression (y-axis) is represented as the percentage of the maximal expression level detected for each staining. (**B–D**) Sialidase treatment elicits expected changes in lectin binding. Tonsil cells were treated with sialidase (20 µg/ml) for 1 hr at 37°C, and then stained with the CyTOF-Lec panel. Shown are histograms depicting the extent of interaction with sialic acid-binding (**B**), fucose-binding, (**C**) or T antigen-binding (**D**) lectins. Removal of sialic acid by sialidase decreases binding by sialic acid-binding lectins, while increasing binding by the fucose- and T antigen-binding lectins, as expected.

The online version of this article includes the following figure supplement(s) for figure 1:

**Figure supplement 1.** Validation of antibodies and lectins used for cytometry by time of flight (CyTOF) analysis.

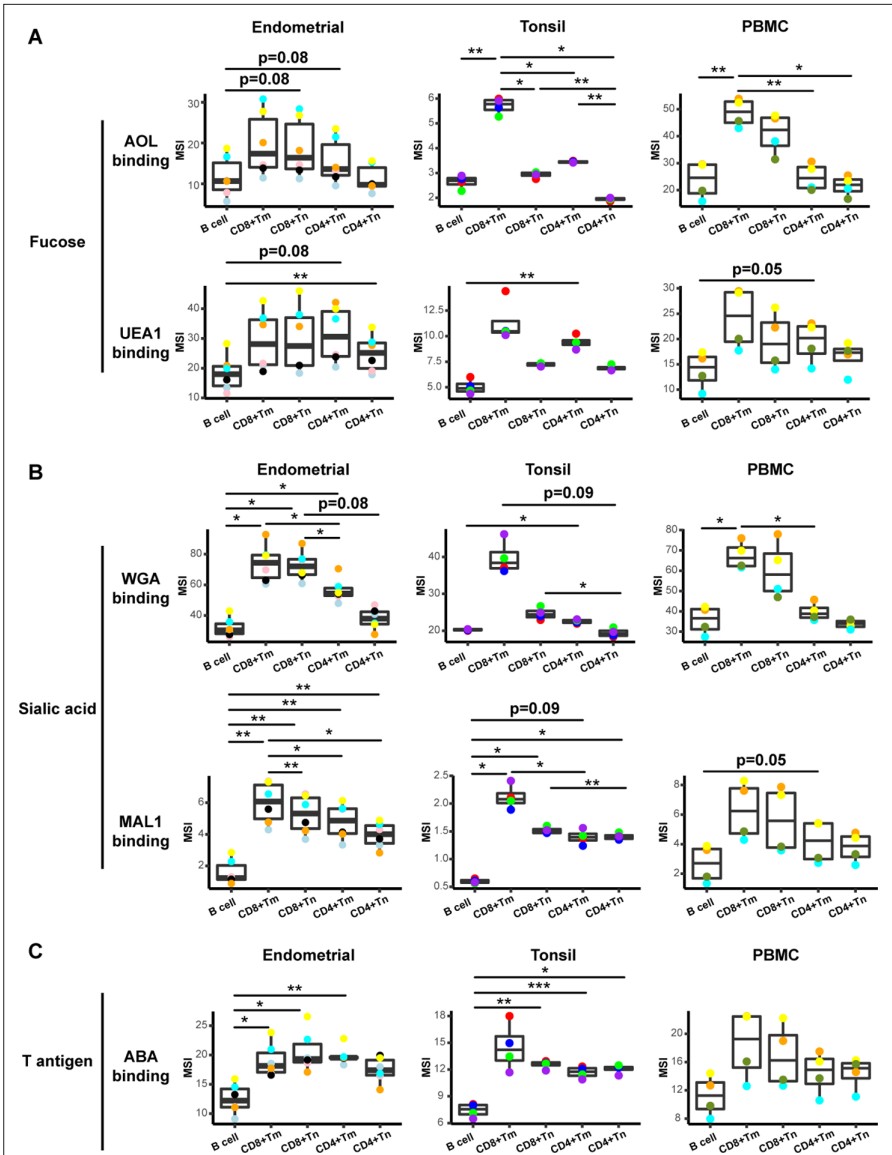

**Figure 2.** Glycan expression in lymphocytes from human endometrium, tonsils, and blood. (**A**) Box plots showing staining by fucose-binding lectins on B and T cells from the endometrium, tonsils, and PBMCs, quantified as median signal intensity (MSI). T cells were subdivided into memory CD8+ T cells (CD8+ Tm), naïve CD8+ T cells (CD8+ Tn), memory CD4+ T cells (CD4+ Tm), and naïve CD4+ T cells (CD4+ Tn) based on their expression of Tm- and Tn-specific CyTOF markers. AOL binds to total/core fucose, UEA1 binds to α1–2 branched fucose. Although there were some differences in binding between sites and between the different lectins, in all instances fucose-binding proteins bound CD4+ Tm at higher levels than they did CD4+ Tn. (**B**) Box plots showing binding by sialic acid-binding lectins WGA and MAL-1. Results are presented as in *panel A*. WGA binds to total sialylated glycans and MAL-1 binds to α2–3 sialylated glycans. Overall, the sialic acid-binding lectins bound CD8+ T cells at higher levels than they did CD4+ T cells and B cells. (**C**) Box plots showing binding by the T antigen-binding lectin ABA. Results are presented as in *panel A*. Overall, ABA bound T cells at higher levels than they did B cells. *p<0.05, **p<0.01, ***p<0.001 as assessed using the Student's paired t test and adjusted for multiple testing using the Holm method.

the known role of core surface fucosylation for T cell activation, which is more prominent within the memory compartment (*Fujii et al., 2016*; *Liang et al., 2018*). The sialic acid-binding lectins WGA and MAL1 also consistently bound to memory T cells more than to their naïve counterparts, although this only reached statistical significance for MAL1 binding of endometrial CD8+ T cells

(*Figure 2B*). Binding by ABA did not show a consistent pattern between memory vs. naïve T cells (*Figure 2C*). Binding by all five lectins was low on B cells, particularly as compared to memory CD8+ T cells (*Figure 2A–C*).

These results establish CyTOF-Lec as a panel that can quantitate glycan and protein expression at the single-cell level, and detect differential glycan expression between different subsets of immune cells.

## HIV alters expression of fucose and sialic acid in a tissue site-dependent manner

We next applied CyTOF-Lec to determine the extent to which glycans are remodeled on the surface of HIV-infected cells. Fresh endometrial biopsies (n=6 donors) and whole tonsils from tonsillectomies (n=4 donors) from HIV seronegative individuals were processed into single-cell suspensions, and then immediately exposed to the HIV-F4.HSA, a replication-competent and Nef-sufficient virus that harbors the CCR5-tropic 109FPB4 transmitted/founder (T/F) Env (*Cavrois et al., 2017*). In addition, PBMCs were isolated from whole blood of HIV seronegative individuals (n=4 donors) and exposed to HIV-F4. HSA. To limit potential confounding effects of ex vivo-induced T cell activation, infection was carried on unstimulated cells. Infection was allowed to proceed for 3 days, after which the cells, as well as cells from paired uninfected control cultures, were harvested for CyTOF-Lec analysis. CD4+ T cells were identified as intact, live singlet CD3+ CD8- cells, while infected cells were identified as intact, live singlet CD3+ CD8- CD4$^{Low}$ cells (*Figure 3—figure supplement 1*), to account for the downregulation of cell-surface CD4 by HIV (*Doms and Trono, 2000*; *Garcia and Miller, 1991*; *Lama, 2003*; *Piguet et al., 1999*). Consistent with our prior studies (*Ma et al., 2020*), endometrial T cells were the most susceptible to HIV-F4.HSA infection (*Figure 3—figure supplement 2A*). HIV-infected cells from all three sites were remodeled, as established qualitatively by assessing their locations on a t-SNE (*Figure 3—figure supplement 2B*) as well as quantitatively using SLIDE (*Sen et al., 2014*; *Figure 3—figure supplement 2C*). The remodeling of the infected cells is consistent with prior studies (*Cavrois et al., 2017*; *Ma et al., 2020*; *Xie et al., 2021*), and confirms that CyTOF-Lec is a valid panel for the analysis of remodeling.

To identify specific glycans that were remodeled, we implemented PP-SLIDE to identify the predicted precursor (PRE) cells (*Cavrois et al., 2017*; *Ma et al., 2020*; *Neidleman et al., 2020b*; *Xie et al., 2021*). PRE cells harbor the predicted original (pre-remodeling) features of T cells infected by HIV and are identified using k-nearest neighbor approaches by matching, in the high-parameter CyTOF space, the T cells in the uninfected culture most similar in phenotype to every HIV-infected cell (*Figure 3—figure supplement 3A*). As expected (*Ma et al., 2020*), the PRE cells from all three sites were preferentially memory CD4+ T cells (*Figure 3—figure supplement 3B, C*). Having identified the PRE cells, we then determined which glycans, if any, were remodeled by HIV infection. This was accomplished by assessing for lectins that differentially bound the PRE as compared to the infected cells. Glycans were quantitated by reporting the median signal intensity (MSI) of their corresponding lectins among each population of cells from each donor. Interestingly, we found that both fucose and sialic acid were upregulated during HIV infection. Infected cells from all three sites potently upregulated total fucose as assessed by AOL binding (*Figure 3A*, *Figure 3—figure supplement 4A*), although this upregulation did not reach statistical significance (after correcting for multiple comparisons) for the endometrium. Binding by UEA1, however, was not significantly different between infected and PRE cells, and tended to be downregulated in the endometrium and tonsils, and upregulated in PBMCs (*Figure 3A*). As UEA1 binds α1–2 branched fucose (*Supplementary file 1*), these results suggest that fucosylation is globally upregulated upon infection of tissue CD4+ T cells with HIV, although not the type of fucosylation that creates α1–2 branched structure. In contrast to the fucose-binding lectins, both sialic acid-binding lectins were increased on infected compared to PRE cells (*Figure 3B*, *Figure 3—figure supplement 4B*), and this was observed for all three sites although results did not reach statistical significance for the endometrium. These results are consistent with an upregulation of both total sialic acid (recognized by WGA, which also binds to *N*-acetylglucosamine [GlcNAc]) (*Schwarz et al., 1999*) and α2–3 linked sialic acid (recognized by MAL-1) by HIV during infection. In contrast to fucose and sialic acid, we did not observe any marked upregulation of T antigen on infected cells (*Figure 3C*).

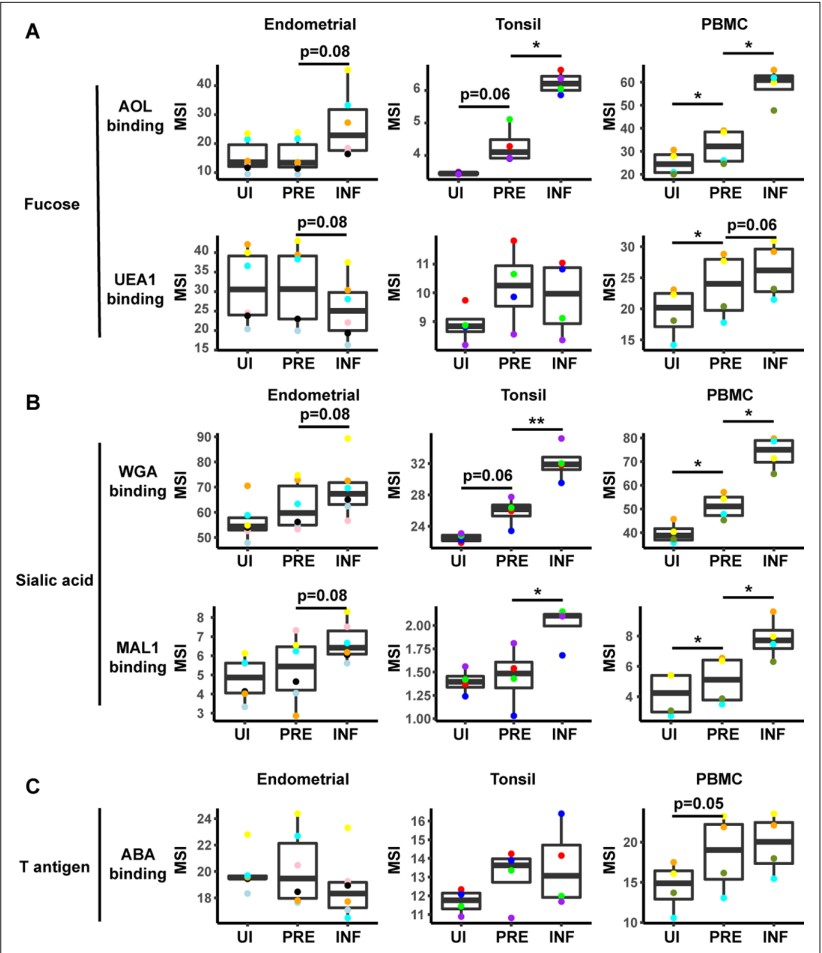

**Figure 3.** HIV alters expression of fucose and sialic acid in a tissue site-dependent manner. (**A**) HIV preferentially infects fucose-expressing cells and further upregulates fucose expression in a tissue site-dependent manner. Box plots showing binding by fucose-binding proteins on uninfected (UI), predicted precursor (PRE), and infected (INF) CD4+ T cells from the endometrium, tonsils, and PBMCs. All populations were pre-gated on live, singlet CD4+ Tm cells. AOL binds total/core fucose, while UEA1 binds α1–2 branched fucose. (**B**) HIV preferentially infects sialic acid-expressing cells and further upregulates sialic acid in a tissue site-dependent manner. Box plots showing binding by sialic acid-binding lectins. Results are presented as in *panel A*. WGA binds total sialylated glycans and MAL1 binds α2–3 sialylated glycans. (**C**) Box plots showing binding by T antigen-binding lectin ABA. Results are presented as in *panel A*. *p<0.05, **p<0.01 as assessed using the Student's paired t test and adjusted for multiple testing using the Benjamini-Hochberg for false discovery rate (FDR).

The online version of this article includes the following figure supplement(s) for figure 3:

**Figure supplement 1.** Cytometry by time of flight (CyTOF) gating strategy to identify uninfected and HIV-infected T cells.

**Figure supplement 2.** HIV remodels T cells from both tissues and blood.

**Figure supplement 3.** HIV preferentially infects memory CD4+ T cells from both tissues and blood.

**Figure supplement 4.** Histogram and t-SNE visualizations of HIV-induced alteration of fucose and sialic acid expression.

**Figure supplement 5.** HIV infection alters expression of fucose and sialic acid in bystander cells.

**Figure supplement 6.** HIV upregulates fucose, sialic acid, and T antigen expression in subsets of bystander cells.

## HIV preferentially infects memory CD4+ T cells with higher fucose and sialic acid levels

In addition to revealing antigens that have been remodeled by infection, the PP-SLIDE approach can also identify antigens that are differentially expressed on cells before infection. In particular, antigens

more abundant on PRE than uninfected cells correspond to antigens preferentially expressed on HIV-susceptible cells, while those less abundant on PRE cells correspond to those preferentially expressed on HIV-resistant cells. We used as our uninfected population CD4+ Tm cells, excluding CD4+ Tn, CD8+ Tm, and CD8+ Tn cells because these latter three populations harbored negligible numbers of HIV-susceptible cells (*Figure 3—figure supplement 3B*). This exclusion was important because otherwise antigens differentially expressed between uninfected and PRE cells could just reflect phenotypic differences between these major subsets. Significant differences in lectin binding between PRE and uninfected (CD4+ Tm) cells were only observed in PBMCs, with AOL, UEA1, WGA, and MAL-1 all binding at significantly higher levels on PRE cells (*Figure 3*, *Figure 3—figure supplement 4C*). Tonsillar PRE cells also bound these lectins more than their uninfected counterparts did, but these results did not reach statistical significance. Endometrial PRE cells did not show significant differences in lectin binding relative to their uninfected counterparts.

These results together with the remodeling analysis suggest that in blood and tonsils (but not endometrium), HIV preferentially infects memory CD4+ T cells with higher levels of fucose and sialic acid, and then further upregulates these cell-surface glycans through viral remodeling.

## HIV infection alters the surface glycome of bystander immune cells in tonsils

Remodeling of cells can occur in a cell-intrinsic manner as a result of direct infection, but may also result from bystander effects. For example, the inflammatory environment elicited by HIV replication may elicit phenotypic changes in bystander (uninfected) cells in the infected culture. We therefore assessed whether HIV infection elicits any glycosylation alterations in bystander cells. To identify bystander memory CD4+ T cells, we gated the infected culture on CD4+ Tm cells that were HSA-negative. Increased binding by all five lectins was observed among bystander tonsillar CD4+ Tm cells relative to their counterparts from uninfected cultures (*Figure 3—figure supplement 5*), suggesting that in at least some tissue sites, remodeling of glycans on bystander CD4+ T cells occurs. Interestingly, however, relative to the bystander CD4+ Tm cells, the infected cells still exhibited higher levels of total fucose and sialic acid (as assessed by AOL and WGA binding, respectively), suggesting possible additional cell-intrinsic glycan remodeling by replicating virus (*Figure 3—figure supplement 5*).

To examine whether HIV alters glycan expression in other bystander cellular subsets, we compared glycan levels on multiple subsets of B and T cells from the uninfected vs. infected cultures (*Figure 3—figure supplement 6*). Only tonsils exhibited significant differences between uninfected vs. bystander cells, and these differences were observed among all subsets. For example, sialic acid levels (as assessed by both WGA and MAL-1 binding) were significantly higher in all the analyzed subsets of bystander B, CD8+, and CD4+ T cells, relative to their counterparts from uninfected cultures. Fucose expression was also uniformly higher among bystander cells, although the difference among CD8+ Tm cells did not reach statistical significance. Differences in levels of T antigen were also observed among B cells, CD8+ Tm cells, and CD4+ Tm cells (*Figure 3—figure supplement 6*). These glycan changes may be elicited by HIV infection-induced inflammatory cytokines (*Breen et al., 1990*; *Contreras et al., 2003*; *Sugawara et al., 2019*), which can alter cell-surface glycosylation patterns (*Dewald et al., 2016*; *Giron et al., 2020a*).

## HIV preferentially infects memory CD4+ T cells from tonsils and PBMCs co-expressing high levels of fucose and sialic acid

The data presented thus far suggest that although there are differences between blood vs. the tissue sites examined, fucose and sialic acid are upregulated on HIV productively infected cells, and CD4+ T cells expressing high levels of fucose or sialic are preferentially targeted for infection. We next conducted manual gating to assess whether the HIV-susceptible cells express high levels of both fucose or sialic acid, or whether they belong to distinct subsets of fucose+ vs. sialic acid+ cells. We focused on the AOL and WGA datasets, as they cover total fucose and different forms of sialic acid, respectively. We first examined, within the HIV-infected cultures, the infection rates among CD4+ Tm cells expressing high vs. low levels of AOL or WGA. In both blood and both tissue compartments, AOL$^{High}$ and WGA$^{High}$ cells exhibited significantly higher HIV infection rates than did AOL$^{Low}$ and WGA$^{Low}$ cells, respectively (*Figure 4A*). To assess the extent to which this high level of infection was due to preferential infection of the AOL$^{High}$ and WGA$^{High}$ CD4+ Tm cells, we next compared, among

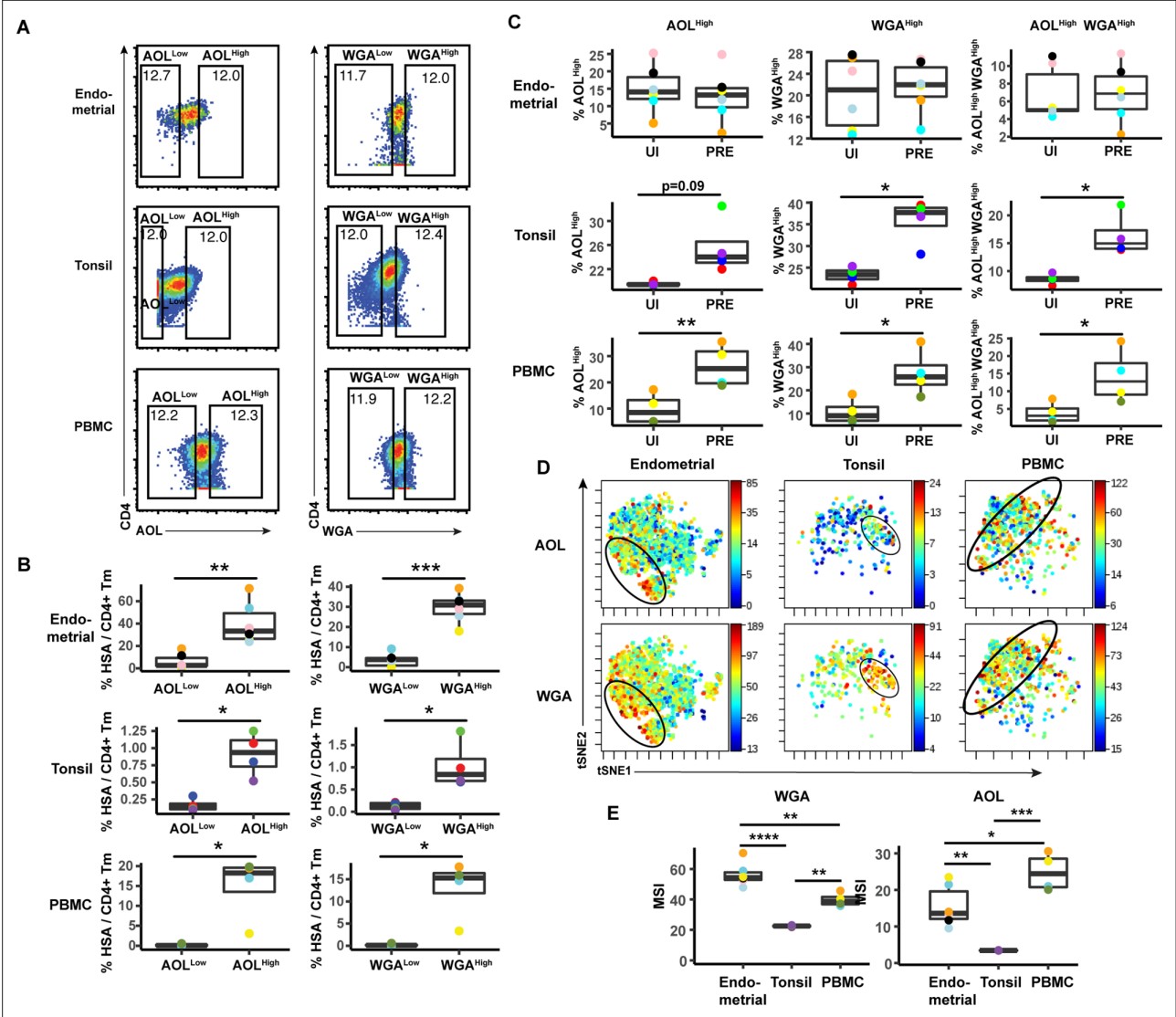

**Figure 4.** HIV preferentially infects memory CD4+ T cells from tonsils and PBMCs with high levels of fucose and sialic acid. (**A**) Gating strategy to identify CD4+ Tm populations expressing different levels of fucose and sialic acid (as detected by AOL and WGA binding, respectively) (**B**) The proportions of CD4+ Tm cells that were productively infected (as assessed by HSA positivity) are higher among the AOL^High^ and WGA^High^ cells than among their AOL^Low^ and WGA^Low^ counterparts for all three sites. *p<0.05, **p<0.01, and ***p<0.001 as assessed using the Student's paired t test. Each color corresponds to a different donor. Gates for the depicted populations are shown in *panel A*. (**C**) Proportion of uninfected CD4+ Tm and PRE cells expressing high levels of fucose, sialic acid, or both (as determined by high binding by AOL or WGA, respectively), as assessed by manual gating. In tonsils and PBMCs, cells expressing fucose and sialic acid were preferentially selected for infection by HIV. *p<0.05, **p<0.01 as assessed using the Student's paired t test. (**D**) Co-expression of fucose and sialic acid on PRE cells in the indicated specimens, as depicted by t-SNE heatmaps. Shown are cells concatenated from all donors analyzed in the study. Regions of the t-SNE co-expressing fucose and sialic acid are circled. (**E**) Levels of fucose and sialic acid differ between CD4+ Tm cells from different origins, as shown by median signal intensity (MSI) for binding by WGA (sialic acid-binding) and AOL (fucose-binding). *p<0.05, **p<0.01, ***p<0.001, ****p<0.0001 as assessed using a one-way ANOVA and adjusted for multiple testing using the Bonferroni.

the uninfected CD4+ Tm cells and PRE cells, the percentages of cells that were AOL^High^, WGA^High^, or AOL^High^ WGA^High^. Consistent with the MSI data, the percentages of cells expressing high levels of AOL or WGA were higher among PRE cells in both tonsils and PBMCs (*Figure 4B*), suggesting preferential infection of fucose- and sialic acid-expressing cells by HIV at these sites. Interestingly, AOL^High^ WGA^High^ cells were also significantly over-represented among PRE cells at these sites (*Figure 4B*), suggesting that the HIV-susceptible Tm cells co-express fucose and sialic acid. Indeed, visualization of the PRE cells by t-SNE revealed cells binding high levels of both AOL and WGA among the tonsillar and blood

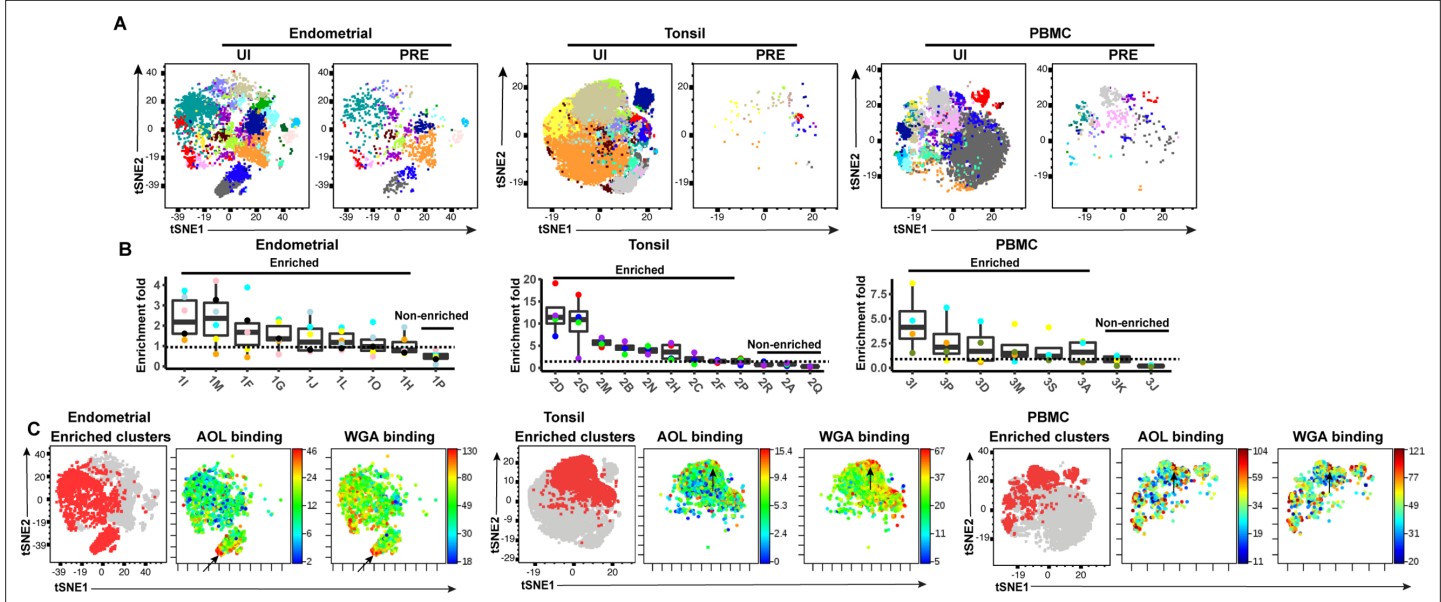

**Figure 5.** FlowSOM clustering confirms that HIV-susceptible subsets from tonsils and PBMCs harbor high levels of fucose and sialic acid. (**A**) t-SNE plots based on FlowSOM analysis of uninfected CD4+ Tm and PRE cells from endometrium, tonsil, and PBMC specimens, showing 20 color-labeled clusters of cells. (**B**) Enrichment of clusters among PRE cells. PRE enrichment-folds were determined by dividing the sizes of each cluster in PRE cells by that in the corresponding uninfected CD4+ Tm cells. Enriched clusters (those with an enrichment fold above 1) correspond to cells preferentially selected for infection. Note that the highest enrichment-folds were observed in tonsils, suggesting the most preferential selection of subsets for infection in this specimen type. Each color corresponds to a different donor. Labels on the x-axis refer to the cluster name. (**C**) Clusters enriched among PRE cells express high levels of fucose and sialic acid, as depicted by t-SNE. For each specimen set, the left-hand t-SNE plot depicts clusters enriched among PRE (*red*) against total cells (*gray*), while the t-SNE plots on the right depict by heatmaps the expression levels of fucose (as assessed by AOL binding) and sialic acid (as assessed by WGA binding) among the enriched clusters. Note that the enriched clusters from all three sites include cells expressing high levels of both fucose and sialic acid (highlighted by arrows).

The online version of this article includes the following figure supplement(s) for figure 5:

**Figure supplement 1.** Levels of glycans on FlowSOM-defined clusters.

compartments (*Figure 4C*). Although PRE cells from the endometrium did not preferentially express fucose or sialic acid (*Figure 4B*), PRE cells co-expressing AOL and WGA could be detected from this site (*Figure 4C*), suggesting the endometrium, like the other two sites, harbors HIV-susceptible cells co-expressing fucose and sialic acid. To better understand why the HIV-susceptible endometrial CD4+ Tm cells, unlike their tonsillar and blood counterparts, did not *preferentially* express high levels sialic acid or fucose, we compared the levels of WGA and AOL binding on uninfected CD4+ Tm cells from the three sites. This analysis revealed WGA, but not AOL, to be expressed at the highest levels on the endometrial cells (*Figure 4D*). These results suggest that the reason WGA[High] cells are not preferentially targeted for infection in the endometrium may be that endometrial CD4+ Tm cells all express high levels of sialic acid. Fucose levels, however, are higher in PBMCs than endometrium, suggesting that the fact that AOL[High] endometrial CD4+ Tm cells aren't preferentially targeted for infection cannot be explained by exceptionally high levels of fucose expression. All together, these results suggest that among blood and tonsillar memory CD4+ T cells, those with the highest levels of fucose and sialic acid are preferentially targeted for infection by HIV; this phenomenon was however not observed among endometrial cells.

We then implemented a more global method of subset identification, using FlowSOM (*Van Gassen et al., 2015*). We combined the uninfected CD4+ Tm and PRE cells from all the donors, and identified 20 clusters for each of the three sites (*Figure 5A*). Endometrial T cells, which had the most PRE cells, were represented among most of the 20 endometrial cell clusters. To determine the extent of enrichment of each cluster among PRE cells, we calculated the ratio of the size of each cluster in the PRE vs. total uninfected CD4+ Tm cells. Enriched clusters identified in this manner (corresponding to those preferentially harboring HIV-susceptible cells, see Materials and methods) were detected among all three sites, with eight clusters from the endometrium, nine from tonsils, and six from PBMCs

(*Figure 5B*). Interestingly, the fold-enrichment was highest among the tonsils (reaching almost 20-fold in one donor), suggesting that of all three sites, this one exhibits the most preferential selection of subsets for infection. Compared to the other two sites, the tonsils also harbored more enriched clusters with significantly elevated levels of fucose, sialic acid, and T antigens relative to their expression levels on uninfected CD4+ Tm cells (*Figure 5—figure supplement 1*). To assess whether the enriched clusters co-express fucose and sialic acid, we assessed by t-SNE heatmaps the levels of AOL and WGA binding on concatenated files of all the enriched clusters from each site. This analysis revealed regions of the t-SNEs co-expressing high levels of fucose and sialic acid (*Figure 5C*), confirming the manual gating data that HIV-susceptible cells co-express these two classes of glycans.

## Total sialylated glycan is a valid marker of highly susceptible CD4+ Tm cells expressing HIV entry receptors and activation markers, and may play a direct role in susceptibility

The results presented thus far suggest that CD4+ Tm cells from endometrium, tonsils, and blood are preferentially susceptible to HIV infection compared to their naïve counterparts, but only in tonsils and blood can high levels of fucose and sialic acid further distinguish HIV-susceptible CD4+ Tm cells from non-susceptible CD4+ Tm cells. To experimentally validate these findings, we conducted sorting experiments. As endometrial and tonsillar T cells do not maintain good viability after sorting, we limited these studies to blood specimens. CD4+ Tm cells from blood expressing low (WGA$^{Low}$), medium (WGA$^{Medium}$), or high (WGA$^{High}$) levels of sialic acid were isolated through sorting (*Figure 6A*). These sorted populations (along with total CD4+ Tm cells as a comparison control) were then exposed to HIV-F4.HSA for 3 days and then assessed by FACS for infection rates. Infection rates directly correlated with the expression levels of sialic acid, with the WGA$^{Low}$ cells being the least susceptible and the WGA$^{High}$ the most (*Figure 6B and C*). These results provide experimental confirmation that in PBMCs, differentially susceptible CD4+ Tm cells can be isolated based solely on sialic acid expression levels.

To better understand the mechanism behind the differential susceptibility of cells expressing high vs. low levels of sialic acid, we returned to our CyTOF datasets and manually gated on CD4+ Tm cells expressing high vs. low levels of total sialic acid as assessed by WGA binding, to assess what was differentially expressed among these two populations. The WGA$^{High}$ cells preferentially expressed fucose (as assessed by AOL binding) (*Figure 6D*), consistent with earlier observations of co-expression of fucose and sialic acid on HIV-susceptible CD4+ T cells. Relative to their WGA$^{Low}$ counterparts, the WGA$^{High}$ cells also preferentially expressed higher levels of CD4 and the HIV co-receptor CCR5 (*Figure 6E*), potentially explaining the increased susceptibility of these cells to infection. As activated T cells are known to be preferentially susceptible to infection (*Stevenson et al., 1990*), we also compared expression levels of activation markers on the WGA$^{Low}$ and WGA$^{High}$ cells, and found that six markers of T cell activation (HLADR, CD69, CD38, CD25, CD28, and ICOS) were all elevated on the latter population (*Figure 6F*). To further validate the notion that the high susceptibility of WGA$^{High}$ CD4+ Tm cells is closely associated with the activation status of these cells, we phenotyped CD4+ Tm cells from resting vs. PHA-stimulated PBMCs. As expected, the stimulated CD4+ Tm cells expressed higher levels of multiple activation markers (*Figure 6—figure supplement 1a*). Importantly, the stimulated CD4+ Tm cells also bound higher levels of WGA, consistent with upregulation of sialic acid upon T cell activation (*Figure 6—figure supplement 1*). Furthermore, activated cells, as defined as those expressing high levels HLADR, CD69, CD38, CD25, CD28, or ICOS, all expressed higher levels of sialic acid as compared to cells with low levels of these activation markers (*Figure 6—figure supplement 1*). These data together strongly support the notion that high sialic expression identifies the most activated subsets of CD4+ Tm cells.

We also considered the possibility that WGA$^{High}$ cells may support higher levels of productive infection because these cells better survive the cytopathic effects of HIV replication. Consistent with this hypothesis, we found that WGA$^{High}$ cells expressed higher levels of CD127, a marker of long-lived self-renewing cells, and Ox40 and BIRC5, which are involved in protecting HIV-infected cells from apoptosis (*Kuo et al., 2018*; *Figure 6G*). In comparison, markers preferentially expressed on the WGA$^{Low}$ cells were those of central memory T cells, including CD27, CCR7, and CD62L (*Figure 6H*). Additional markers in our CyTOF panel differentially expressed between the WGA$^{Low}$ and WGA$^{High}$ cells included

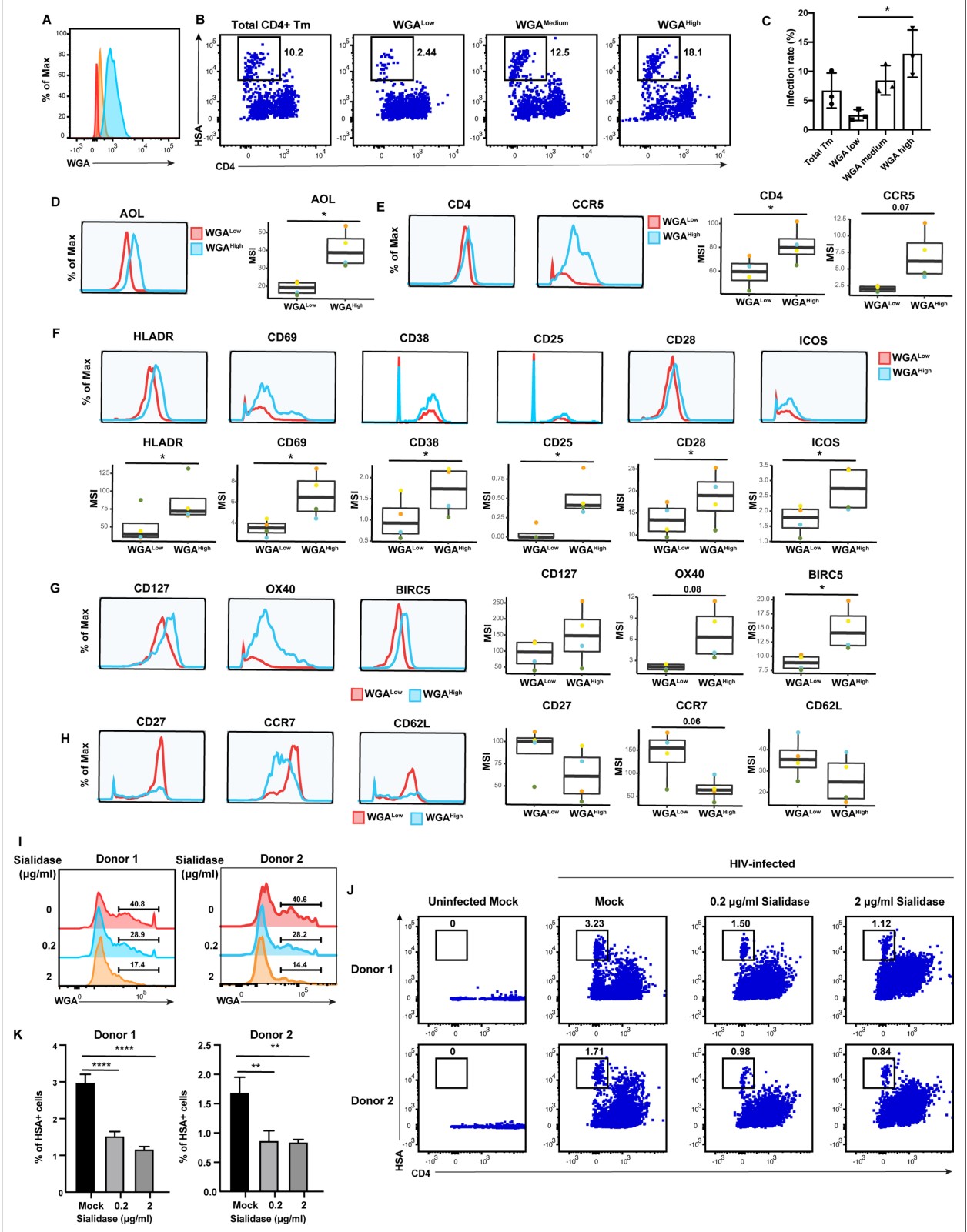

**Figure 6.** High levels of sialylated glycans identifies highly susceptible and activated CD4+ Tm cells, and plays a direct role in susceptibility. (**A**) Histograms showing the expression of total sialylated glycans on three populations (WGA^Low [red], WGA^Medium [yellow], and WGA^High [blue]) of sorted uninfected CD4+ Tm cells (CD3+ CD4+ CD45RA-), as assessed by WGA binding. One of three representative donors is shown. (**B–C**) The sorted uninfected CD4+ Tm cells in *panel A*, along with total CD4+ Tm cells, were exposed to F4.HSA and assessed by flow cytometry for infection rates 3 days

*Figure 6 continued on next page*

*Figure 6 continued*

later. Results are gated on live, singlet CD3+ CD8- cells. Shown are representative FACS plots from one donor (**B**) and compiled results from three donors (**C**). For each donor, experimental duplicates were performed for each condition. Each datapoint shown corresponds to one donor. *p<0.05 as assessed using a one-way ANOVA and adjusted for multiple testing using the Bonferroni. (**D**) WGA$^{High}$ Tm cells bind more AOL than WGA$^{Low}$ Tm cells do. Shown are the histogram plots from one representative PBMC donor (*left*) and box plots from all four PBMC donors (*right*). (**E**) WGA$^{High}$ Tm cells express more CD4 and CCR5 than WGA$^{Low}$ Tm cells do. Shown are the histogram plots from one representative PBMC donor (*left*) and box plots from all four PBMC donors (*right*). (**F**) WGA$^{High}$ Tm cells express higher levels of activation markers than WGA$^{Low}$ Tm cells do. Shown are the histogram plots from one representative PBMC donor (*top*) and box plots from all four PBMC donors (*bottom*). (**G**) WGA$^{High}$ Tm cells express higher levels of the pro-survival factors CD127, BIRC5, and Ox40 than WGA$^{Low}$ Tm cells do. Shown are the histogram plots from one representative PBMC donor (*left*) and box plots from all four PBMC donors (*right*). (**H**) The CD127, CCR7, and CD62L receptors are expressed at lower levels in WGA$^{High}$ relative to WGA$^{Low}$ Tm cells. Shown are the histogram plots from one representative PBMC donor (*left*) and box plots from all four PBMC donors (*right*). For panels D–G, *p<0.05 as assessed using the Student's paired t test and adjusted for multiple testing using the Benjamini-Hochberg for false discovery rate (FDR). (**I**) Transient treatment with sialidase decreases cell-surface levels of sialidase on CD4+ T cells. PBMCs were treated for 1 hr with sialidase prior to assessment of cell-surface WGA binding. Shown are overlaid histograms demonstrating a decrease in cell-surface sialic acid levels (as reflected by WGA binding) in the sialidase-treated cells from two independent donors. Results are gated on live, singlet CD3+ CD8- CD4+ cells. Numbers correspond to percent of cells within the indicated gate. (**J–K**) PBMCs treated for 1 hr with the indicated concentrations of sialidase were exposed to F4.HSA and assessed by flow cytometry for infection rates 3 days later. Results are gated on live, singlet CD3+ CD8- cells. Shown are representative FACS plots from two donors (**J**) and the results of experimental triplicates from each of these donors (**K**). **p<0.01 and ****p<0.0001 as assessed using a one-way ANOVA and adjusted for multiple testing using the Bonferroni.

The online version of this article includes the following figure supplement(s) for figure 6:

**Figure supplement 1.** Activated CD4+ Tm cells express high levels of sialic acid.

**Figure supplement 2.** Expression levels of cytometry by time of flight (CyTOF) antigens on WGA$^{Low}$ and WGA$^{High}$ Tm cells.

other glycans, transcription factors, homing receptors, and exhaustion markers (*Figure 6—figure supplement 2*).

Although these results suggest that high levels of sialic acid may simply be a biomarker of the most HIV-susceptible CD4+ Tm cells, it did not rule out the possibility that sialic acid plays a direct role in HIV susceptibility. To test this, we tried two approaches to diminish cell-surface sialic acid expression: transient treatment with either sialic acid synthase inhibitor P-3F$_{AX}$-Neu5Ac or sialidase. Inhibitor treatment did not decrease cell-surface sialic acid levels (*Figure 6—figure supplement 1*), but sialidase did (*Figure 6I*). We therefore exposed mock- or sialidase-treated cells to HIV-F4.HSA for 3 days and then assessed infection rates by FACS. We found that infection rates were decreased in a sialidase dose-dependent manner (*Figure 6J and K*). These results suggest that a single glycan – sialic acid – may not only be capable of distinguishing memory CD4+ T cells with vastly different phenotypic features and HIV susceptibility but may also play a direct role in promoting infection by the virus.

## Discussion

We describe here the development and implementation of CyTOF-Lec, a high-parameter single-cell method to simultaneously quantitate multiple glycans and proteins on the surface of human cells. We used CyTOF-Lec to identify glycomic and phenotypic differences between immune subsets from blood and multiple tissues, and between uninfected and HIV-infected CD4+ T cells. Moreover, by performing PP-SLIDE bioinformatics on our datasets of HIV-infected blood and tissue cells, we identified unique glycan features characteristic of HIV-susceptible CD4+ T cells, and identified glycan structures that were remodeled as a result of cell-intrinsic HIV replication.

Multiple studies have demonstrated remodeling of host cells upon HIV infection. HIV infection of cell lines (*Matheson et al., 2015*) as well as primary cells (*Cavrois et al., 2017*; *Ma et al., 2020*; *Xie et al., 2021*) leads to up- and downregulation of a variety of host proteins, reflecting the ability of the virus to hijack host processes to complete its replication cycle. We demonstrate in the current study that HIV also remodels the surface glycome of infected cells. By implementing PP-SLIDE to discern antigens differentially expressed as a result of remodeling from those differentially expressed as a reflection of HIV selection, we found that both fucose and sialic acid were upregulated on CD4+ T cells after infection. As core fucosylation on the cell surface is critical for CD4+ T cell activation (*Fujii et al., 2016*; *Liang et al., 2018*) and activated cells are more permissive to HIV infection (*Stevenson et al., 1990*), HIV may upregulate fucosylation to maintain an elevated state of activation facilitating the completion of its replication. We cannot rule out, however, that upregulation of fucose is

a byproduct of T cell activation, and not a direct consequence of HIV infection. Of note, fucosylation is also important for lymphocyte trafficking (*Colomb et al., 2020*), and the upregulation of fucose on infected cells, even if not directly promoting viral replication, may facilitate the circulation of infected cells from blood into tissues where ample populations of HIV-permissive CD4+ T cells reside.

Sialic acid, like fucose, was also upregulated as a result of HIV infection. In general, sialylated glycans on the surface of cells can elicit an immunosuppressive response by effector cells of the immune system. For example, binding of sialic acid on the surface of cancer cells to Siglec-7 and Siglec-9 proteins on the surface of NK cells can diminish NKG2D-mediated activation of the NK cells (*Xiao et al., 2016*). Moreover, NKG2D may play an important role in NK cell-mediated killing of HIV-infected cells and NKG2D ligands are actively downregulated by HIV Nef to escape immune recognition (*Jost and Altfeld, 2012*). We postulate that through a combination of upregulating sialic acid and downregulating NKG2D ligands, HIV may evade NKG2D-mediated killing by NK cells.

Interestingly, sialic acid was also preferentially expressed on the memory CD4+ T cells that were preferentially targeted for infection by HIV. These results suggest that the high levels of sialic acid on HIV-infected cells results from preferential infection of cellular subsets with high levels of cell-surface sialic acid, followed by further upregulation of this glycan. Interestingly, however, the preferential selection of WGA$^{High}$ memory CD4+ T cells for HIV infection was only observed in tonsils and PBMCs, but not in the endometrium (*Figure 4C*). Because memory CD4+ T cells from the endometrium express higher levels of cell-surface sialic acid than those from tonsils or PBMCs (*Figure 4E*), most endometrial memory CD4+ T cells may already have the WGA$^{High}$ HIV-permissive phenotype, resulting in minimal selection of these cells by HIV. Indeed, this would be consistent with the many subsets of endometrial memory CD4+ T cells that are susceptible to HIV infection, as compared to a markedly lower number of HIV-susceptible memory CD4+ T cell subsets from the blood compartment (*Ma et al., 2020*). It is also worthwhile to note that the endometrium is a unique tissue in that it serves as the site of blastocyst implantation and development of a semi-allogenic fetus. The unusually high levels of sialic acid on immune cells from this tissue may be important to prevent immune-mediated rejection of spermatozoa or the developing fetus.

Importantly, HIV's preference for cells expressing high levels of sialic acid was validated by demonstrating via sorting experiments that blood-derived memory CD4+ T cells expressing high levels of WGA were significantly more susceptible to infection than those expressing low levels of WGA. The WGA$^{High}$ population exhibits many features that could explain its high susceptibility: higher expression levels of HIV receptor/co-receptor, higher activation status, and higher expression of markers associated with survival of HIV-infected cells. However, our somewhat surprising finding that sialidase-treated cells are more poorly susceptible to HIV additionally suggest that sialic acid may exert a direct role in promoting HIV infection; the precise step in the viral replication cycle affected by sialic acid should be interrogated in future studies. Regardless, since thus far markers that distinguish memory CD4+ T cells with differing susceptibilities to HIV have been difficult to identify, our findings that WGA may be useful as a tool to isolate a highly susceptible population of CD4+ T cells could be of practical use for the field. On a more fundamental level, our results suggest that host glycan expression can very much influence HIV replication dynamics.

Of note, we cannot rule out the possibility that the high levels of fucose and sialic acid on HIV-infected cells may also result in part from binding of HIV virions to the surface of the infected CD4+ T cells. HIV particles contain cell-derived glycolipids, including the sialic acid-containing GM3 ganglioside (*Puryear et al., 2012*). Furthermore, HIV gp120 is heavily N-glycosylated, with a majority of high-mannose N-glycan structures and a lower proportion of complex N-glycans carrying lactosamine residues and terminal sialic acid. Sialic acid on gp120 is recognized by multiple members of the Siglec family of lectins, and these lectins can facilitate *trans-* or cell-to-cell infection of CD4+ T cells (*Izquierdo-Useros et al., 2012*; *Varchetta et al., 2013*; *Zou et al., 2011*). However, we note that our studies characterizing infected cells defined these cells as those that express HSA – which is not on the surface of input virions – and that have downregulated cell-surface CD4, a hallmark of productive HIV infection. The fact that we found unique glycan features of these cells relative to bystander CD4+ T cells suggests that productively infected cells have distinct cell-surface glycan features in a manner not attributable to just surface virion sticking.

Our study has limitations. First, since cells from the different compartments (blood, tonsils, and endometrium) were run in separate batches, it is difficult to compare expression levels between cells

from these compartments, which is why our analysis focused on relative glycan expression levels among cells within each compartment. Second, while WGA is commonly used as a reagent to monitor cell-surface sialic acid levels, it also binds GlcNAc, so we cannot rule out that some of the WGA effects we observed were due to GlcNAc and not sialic acid. Notably, a recent study reported that cell-surface GlcNAc is important for HIV binding and infection (*Spillings et al., 2022*), which is in line with our observation that WGA^High cells are highly susceptible to HIV infection. Third, a limited number of donors were analyzed, particularly in the sialidase treatment experiments, as those experiments were technically challenging due to the toxic effects of sialidase treatment on primary CD4+ T cells from most tested donors.

While our study focused on characterizing cells productively infected with HIV-1, identified as those expressing on their cell surface the LTR-driven HSA reporter protein, we envision that CyTOF-Lec will also be useful for studying the HIV reservoir that persists despite antiretroviral therapy (ART). Recent observations suggest that a significant portion of the reservoir in ART-suppressed people living with HIV (PLWH) is transcriptionally active, expressing short and incomplete HIV transcripts in the absence of ex vivo stimulation (*Yukl et al., 2018*). Interestingly, we recently demonstrated using cells from ART-suppressed PLWH that CD4+ T cells expressing high levels of fucose contain significantly more transcriptionally active HIV reservoir cells than those expressing low levels of fucose (*Colomb et al., 2020*). With the development of CyTOF-Lec as a single-cell analysis tool, we will now be able to interrogate glycan expression patterns on transcriptionally active, as well as transcriptionally silent HIV reservoir cells. Specifically, we envision combining CyTOF-Lec with PP-SLIDE analysis of reservoir cells from PLWH. This will entail using bioinformatics approaches to trace ex vivo reactivated cells to their original pre-stimulation states by PP-SLIDE, an approach we previously described and validated experimentally (*Neidleman et al., 2020b*), but under conditions where we incorporate assessment of the glycan features of the reservoir cells with the CyTOF-Lec panel.

Beyond HIV, CyTOF-Lec has potentially wide-ranging applications. Glycosylated host receptors play important roles in enabling the attachment of virus. For example, influenza virus enters cells by binding of its hemagglutinin (HA) protein to terminal sialic acid moieties attached to surface proteins of host cells (*Li et al., 2021*). The novel beta-coronavirus SARS-CoV-2 may also exploit host cell glycosylation for infection, as its primary receptor ACE2 is heavily glycosylated. Indeed, blockage of N- and O-glycan synthesis on the host cell can inhibit SARS-CoV-2 entry by diminishing the ability of the viral spike protein to bind ACE2 (*Yang et al., 2020*). Better understanding the surface glycans of virally susceptible host cells through single-cell glycomic analysis by CyTOF-Lec, in conjunction with identifying the protein backbones harboring the glycans, will improve our understanding of viral pathogenesis and can potentially lead to novel prophylactic or therapeutic agents for clinically relevant viral pathogens.

Further development of CyTOF-Lec in conjunction with next-generation sequencing approaches will also be valuable for developing the tools to better understand how glycan expression associates with or directly regulates host cell processes. A recent report combined glycomic analysis with single-cell RNAseq (*Kearney et al., 2021*) by incubating cells with a biotinylated version of the lectin L-Pha followed by a DNA-barcoded anti-biotin antibody. Subsequent droplet encapsulation and sequencing revealed the transcriptomes of cells with different levels of L-Pha binding. We envision that by directly conjugating lectins to barcoded DNA oligos, we can expand the numbers of lectins we can simultaneously monitor. Furthermore, additional inclusion of DNA-barcoded antibodies (*Peterson et al., 2017*; *Stoeckius et al., 2017*) will enable simultaneous surface proteome analysis. Such a technology, while not as high-throughput, scalable, and cost-effective as CyTOF-Lec, will enable a genome-wide analysis of cellular processes associated with differential surface glycosylation.

# Materials and methods

**Key resources table**

| Reagent type (species) or resource | Designation | Source or reference | Identifiers | Additional information |
|---|---|---|---|---|
| Strain, strain background (*Escherichia coli*) | Stbl3 | Fisher | C737303 | |

*Continued on next page*

Continued

| Reagent type (species) or resource | Designation | Source or reference | Identifiers | Additional information |
|---|---|---|---|---|
| Cell line (human) | 293T cells | ATCC | CRL-3216 | |
| Biological sample (human) | Endometrial tissue | Women's Health Clinic of Naval Medical Center Portsmouth (NMCP) | | |
| Biological sample (human) | Tonsil | Cooperative Human Tissue Network (CHTN) | | |
| Biological sample (human) | Blood | Vitalant Research Institute and Stanford Blood Bank | | |
| Peptide, recombinant protein | AOL | TCI Chemicals | Cat# L0169 | CyTOF (1:20) |
| Antibody | Anti-Human CD49d (9F10) (Mouse, Monoclonal) | Fluidigm | Cat# 3141004B | CyTOF (1:200) |
| Antibody | Anti-Human CD195/CCR5 (NP-6G4) (Mouse, Monoclonal) | Fluidigm | Cat# 3144007A | CyTOF (1:50) |
| Antibody | Anti-Human CD8 (RPA-T8) (Mouse, Monoclonal) | Fluidigm | Cat# 3146001B | CyTOF (1:33) |
| Antibody | Anti-Human CD7 (CD7-6B7) (Mouse, Monoclonal) | Fluidigm | Cat# 3147006B | CyTOF (1:200) |
| Antibody | Anti-CD278/ICOS (C398.4A) (Mouse, Monoclonal) | Fluidigm | Cat# 3148019B | CyTOF (1:100) |
| Antibody | Anti-Mouse CD24 (M1/69) (Rat, Monoclonal) | Fluidigm | Cat# 3150009B | CyTOF (1:200) |
| Peptide, recombinant protein | MAL-1 | Vector Laboratories | Cat# L-1310–5 | CyTOF (1:25) |
| Peptide, recombinant protein | WGA | Vector Laboratories | Cat# L-1020–25 | CyTOF (1:15) |
| Antibody | Anti-Human CD62L (DREG-56) (Mouse, Monoclonal) | Fluidigm | Cat# 3153004B | CyTOF (1:50) |
| Antibody | Anti-Human TIGIT (MBSA43) (Mouse, Monoclonal) | Fluidigm | Cat# 3154016B | CyTOF (1:15) |
| Antibody | Anti-Human CD196 (CCR6) (Mouse, Monoclonal) | BD Biosciences | Cat# 559560 | CyTOF (1:25) |
| Peptide, recombinant protein | UEA-1 | Vector Laboratories | Cat# L-1060–5 | CyTOF (1:33) |
| Antibody | Anti-Human CD134/OX40 (ACT35) (Mouse, Monoclonal) | Fluidigm | Cat# 3158012B | CyTOF (1:25) |
| Antibody | Anti-Human CD197/CCR7 (G043H7)- (Mouse, Monoclonal) | Fluidigm | Cat# 3159003A | CyTOF (1:25) |
| Antibody | Anti-Human CD28 (CD28.2) (Mouse, Monoclonal) | Fluidigm | Cat# 3160003B | CyTOF (1:100) |
| Antibody | Anti-human CD45RO (Mouse, Monoclonal) | Biolegend | Cat# 304239 | CyTOF (1:25) |
| Antibody | Anti-Human CD69 (FN50) (Mouse, Monoclonal) | Fluidigm | Cat# 3162001B | CyTOF (1:400) |
| Antibody | Anti-Human CD294/CRTH2 (BM16) (Rat, Monoclonal) | Fluidigm | Cat# 3163003B | CyTOF (1:50) |
| Antibody | Anti-Human CD279 (PD-1) (Mouse, Monoclonal) | BD Biosciences | Cat# 562138 | CyTOF (1:50) |
| Antibody | Anti-Human CD127/IL-7Ra (A019D5) (Mouse, Monoclonal) | Fluidigm | Cat# 3165008B | CyTOF (1:25) |

Continued

| Reagent type (species) or resource | Designation | Source or reference | Identifiers | Additional information |
|---|---|---|---|---|
| Antibody | Anti-Human CXCR5 (CD185) (Rat, Monoclonal) | BD Biosciences | Cat# 552032 | CyTOF (1:50) |
| Antibody | Anti-Human CD27 (L128) (Mouse, Monoclonal) | Fluidigm | Cat# 3167006B | CyTOF (1:100) |
| Antibody | Anti-Human CD30 (Mouse, Monoclonal) | BD Biosciences | Cat# 555827 | CyTOF (1:33) |
| Antibody | Anti-Human CD45RA (HI100) (Mouse, Monoclonal) | Fluidigm | Cat# HI100 | CyTOF (1:200) |
| Antibody | Anti-Human CD3 (UCHT1) (Mouse, Monoclonal) | Fluidigm | Cat# 3170001B | CyTOF (1:100) |
| Peptide, recombinant protein | ABA | Vector Laboratories | Cat# L-1420–2 | CyTOF (1:33) |
| Antibody | Anti-Human CD38 (HIT2) (Mouse, Monoclonal) | Fluidigm | Cat# 3172007B | CyTOF (1:200) |
| Antibody | α4β7 | Gift from E Butcher | | CyTOF (1:100) |
| Antibody | Anti-Human CD4 (SK3) (Mouse, Monoclonal) | Fluidigm | Cat# 3174004B | CyTOF (1:50) |
| Antibody | Anti-Human CD184/CXCR4 (12G5) (Mouse, Monoclonal) | Fluidigm | Cat# 3175001B | CyTOF (1:50) |
| Antibody | Anti-Human CD25 (Mouse, Monoclonal) | BD Biosciences | Cat# 555430 | CyTOF (1:300) |
| Antibody | Anti-human/mouse Cutaneous Lymphocyte Antigen (CLA) Antibody (Rat, Monoclonal) | Biolegend | Cat# 321302 | CyTOF (1:50) |
| Antibody | HLA-DR Monoclonal Antibody (TU36), Qdot 655 (Mouse, Monoclonal) | Thermo Fisher | Cat# Q22158 | CyTOF (1:50) |
| Antibody | ROR gamma (t) (Rat, Monoclonal) | Thermo Fisher | Cat# 14-6988-82 | CyTOF (1:25) |
| Antibody | Anti-Human NFAT1 (Rat, Monoclonal) | Fluidigm | Cat# 3143023A | CyTOF (1:100) |
| Antibody | Human Survivin Antibody (Mouse, Monoclonal) | R&D Systems | Cat# MAB886 | CyTOF (1:15) |
| Antibody | T-bet Monoclonal Antibody (Mouse, Monoclonal) | Thermo Fisher | Cat# 14-5825-82 | CyTOF (1:25) |
| Antibody | CD152 (CTLA-4) Monoclonal Antibody (Mouse, Monoclonal) | Thermo Fisher | Cat# 14-1529-82 | CyTOF (1:50) |
| Commercial assay or kit | Lenti-X p24$^{Gag}$ Rapid Titer Kit | Takara | Cat# 632200 | |
| Commercial assay or kit | EasySep CD4 enrichment kit | Stem Cell Technologies | | |
| Commercial assay or kit | CD45RA MicroBeads, human | Miltenyi Biotec | | |
| Commercial assay or kit | Zombie Aqua Fixable Viability Kit | Biolegend | Cat# 423102 | FACS (1:100) |
| Antibody | APC/Cyanine7 anti-human CD3 Antibody (Mouse, Monoclonal) | Biolegend | Cat# 344818 | FACS (1:100) |
| Antibody | PE/Cyanine7 anti-human CD4 Antibody (Rat, Monoclonal) | Biolegend | Cat# 357410 | FACS (1:100) |
| Antibody | APC anti-human CD8 Antibody (Mouse, Monoclonal) | Biolegend | Cat# 344722 | FACS (1:100) |

*Continued*

| Reagent type (species) or resource | Designation | Source or reference | Identifiers | Additional information |
|---|---|---|---|---|
| Antibody | FITC Rat Anti-Mouse CD24 (Rat, Monoclonal) | BD Biosciences | Cat# 561777 | FACS (1:100) |
| Software, algorithm | CyTOF software (6.7.1014) | Fluidigm | | |
| Software, algorithm | FlowJo software (10.7.2) | FlowJo LLC, BD Biosciences | | |
| Software, algorithm | Cytobank (9.1, 2022) | Cytobank, Inc. | | |

## Tissue processing and cell isolation

Endometrial tissues were obtained from the Women's Health Clinic of Naval Medical Center Portsmouth (NMCP) in Virginia (CIP # NMCP.2016.0068) under standard operating procedures (*Fassbender et al., 2014*). The biopsies were transported to San Francisco in MCDB-105 (Sigma-Aldrich M6395) containing 10% heat-inactivated fetal bovine serum (FBS) and 1% penicillin/streptomycin (P/S). The collection of endometrial T cells was performed similar to recently described methods (*Ma et al., 2020*). Briefly, endometrial tissue was washed once with SCM media, which consisted of 75% phenol red-free Dulbecco's Modified Eagle's Medium (DMEM, Life Technologies), 25% MCDB-105, 10% FBS, 1% L-glutamine with P/S (Gemini), 1 mM sodium pyruvate (Sigma-Aldrich), and 5 mg/ml insulin (Sigma-Aldrich). The tissues were then digested at 37°C for 2 hr under rotation in SCM media mixed at a 1:1 dilution with Digestion Media, which consisted of HBSS containing $Ca^{2+}$ and $Mg^{2+}$ supplemented with 3.4 mg/ml collagenase type 1 (Worthington Biochemical Corporation LS004196) and 100 U/ml hyaluronidase (Sigma-Aldrich H3631). Cells were then filtered through a Falcon 40 µm cell strainer. The filtrate was then centrifuged, washed once with R10 media (RPMI 1640 containing 10% FBS and 1% P/S), and the cells were cultured in 96-well U-bottom polystyrene plates at a concentration of $10^6$ cells/well in 200 µl R10 media.

## Processing of human lymphoid aggregate cultures

Human tonsils obtained from the Cooperative Human Tissue Network (CHTN) were processed similarly to methods recently described (*Ma et al., 2020*). Briefly, tonsils were rinsed in tonsil media (RPMI supplemented with 15% FBS, 100 µg/ml gentamicin, 200 µg/ml ampicillin, 1 mM sodium pyruvate, 1% non-essential amino acids [Mediatech], 1% Glutamax [Thermo Fisher], and 1% Fungizone [Invitrogen]), dissected into small pieces, and then pressed through a 40 µm cell strainer using a syringe plunger. The cells were then filtered through a second 40 µm cell strainer, centrifuged, and resuspended with 200 µl tonsil media per $10^6$ cells. For sialidase treatment, $10^6$ tonsil cells were resuspended in 1 ml of tonsil media, and then treated with 20 µg of the sialidase for 1 hr at 37°C. The cells were washed once with PBS and processed for CyTOF analysis as described further below.

## PBMC isolation

PBMCs were isolated from reduction chambers obtained from Vitalant Research Institute and Stanford Blood Bank using Ficoll-Hypaque density gradients, and then cultured in R10. For sorting experiments, CD4+ T cells were purified by negative selection using the EasySep CD4 enrichment kit (Stem Cell Technologies), and further enriched for memory cells by depletion of naïve T cells using CD45RA beads (Miltenyi Biotec), prior to lectin staining and sorting as described further below. Where indicated, PBMCs were first stimulated for 3 days with 5 µg/ml PHA in the presence of 10 IU/ml IL-2 prior to CyTOF-Lec analysis.

## Virus preparation and infection assays

Viral stocks of the previously described HIV-1 reporter virus F4.HSA (*Cavrois et al., 2017*) were prepared similarly to recently described methods (*Ma et al., 2020*). Briefly, 293T cells (purchased directly from ATCC and therefore assumed to be authenticated, tested negative for mycoplasma contamination) were seeded in T175 flasks and transfected using polyethylenimine (Polysciences) with F4.HSA proviral DNA (70 µg/flask) (*Longo et al., 2013*). Two days after transfection, supernatants from 293T cultures were harvested, filtered through a 0.22 µm filter, and concentrated by ultracentrifugation at 20,000 rpm (Beckman Coulter Optima XE-90) for 2 hr at 4°C. p24$^{Gag}$ concentrations were

quantitated using the Lenti-X p24$^{Gag}$ Rapid Titer Kit (Clontech). For infection, 10–20 ng/ml p24$^{Gag}$ of F4.HSA was incubated with $10^6$ cells in 200 µl R10 media in 96-well U-bottom polystyrene plates. After 2 hr, cells were fed with fresh R10 media and cultured for another 3 days. Where indicated, PBMCs were first treated with the sialic acid inhibitor P-3F$_{AX}$-Neu5Ac (Tocris) for 24 hr, or sialidase for 1 hr, prior to infection with F4.HSA.

## Flow cytometry

For sorting experiments, purified memory CD4+ cells (see above) were washed once with FACS buffer in 96-well V-bottom polystyrene plates, and then stained at room temperature for 15 min with a 1:200 dilution of the LIVE/DEAD Zombie Aqua Fixable Viability Kit (Biolegend) reagent to exclude dead cells. The cells were washed and stained for 30 min at 4°C with 5 µg/ml FITC-WGA (Vector Laboratories). After the wash, the purified memory CD4 + T cells were resuspended at a concentration of $10^6$ cells/ml, and sorted on an Aria II flow cytometer (BD Biosciences) into WGA$^{Low}$, WGA$^{Medium}$, and WGA$^{High}$ populations. Total CD4+ Tm cells were also sorted as a control. The purity of the sorted cells was confirmed by analysis on Aria II immediately after sorting. All sorted populations were infected with F4.HSA for 3 days. For FACS analysis of the samples, 0.1–1 million cells of each sample were transferred into 96-well V-bottom polystyrene plates, washed once with FACS buffer (PBS containing 2% FBS and 2 mM EDTA), and stained for 30 min on ice with an antibody cocktail consisting of APC/ Cy7-CD3 (SK7, Biolegend), PE/Cy7-CD4 (A161A1, Biolegend), APC-CD8 (SK1, Biolegend), FITC-CD24 (HSA, M1/69, BD Biosciences), and the LIVE/DEAD Zombie Aqua Fixable Viability Kit (Biolegend). The cells were then washed twice, fixed with 1% PFA (Electron Microscopy Sciences) in PBS, and analyzed by flow cytometry on an LSRFortessa (BD Biosciences).

## CyTOF data generation

A 39-parameter CyTOF panel was designed for this study, which included antibodies against markers of T cell differentiation states, activation markers, transcription factors, and homing receptors, and an antibody against HSA to identify productively infected cells. The panel also included numerous lectins enabling the characterization of glycan features (*Supplementary file 1*). X8 antibody-labeling kits (Fluidigm) were used to label antibodies that required in-house conjugation. The conjugated antibodies were quantitated for protein content by Nanodrop (Thermo Fisher). Prior to storage at 4°C, specimens were diluted 1:1 using a PBS-based Antibody Stabilizer (Boca Scientific) supplemented with 0.05% sodium azide.

Preparation of samples for CyTOF staining was conducted as previously described (*Ma et al., 2020*; *Neidleman et al., 2020a*; *Neidleman et al., 2020b*), Briefly, 1–6 million cells were washed once with CyFACS (metal contaminant-free PBS [Rockland] supplemented with 0.1% bovine serum albumin and 0.1% sodium azide). Where indicated, cells were first treated with 20 µg/ml of sialidase or PBS as control and incubated for 1 hr at 37°C. Sialidase was prepared in-house using the *Vibrio cholerae nanH* gene cloned into the pCVD364 vector, which was provided by Dr Eric R Vimr from the University of Illinois Urbana (*Taylor et al., 1992*). After centrifugation, the cells were resuspended with contaminant-free PBS (Rockland) supplemented with 2 mM EDTA (PBS/EDTA), and then treated with 25 µM cisplatin (Sigma-Aldrich) in 4 ml PBS/EDTA for 60 s at room temperature. The samples were then immediately quenched with CyFACS, centrifuged, resuspended in 2% PFA in CyFACS, and incubated for 10 min at room temperature. The cells were then washed three times with CyFACS, resuspended in 100 µl of CyFACS containing 10% DMSO, and frozen at –80°C until CyTOF staining.

To stain multiple specimens in the same reaction, cells were barcoded using the Cell-ID 20-Plex Pd Barcoding Kit according to the manufacturer's instructions (Fluidigm). Briefly, 1–3 million cisplatin-treated cells were thawed and transferred into Nunc 96 DeepWell polystyrene plates (Thermo Fisher). After two washes with Barcode Perm buffer (Fluidigm), the cells were incubated with selected barcodes for 30 min. Cells were then washed with 0.8 ml Maxpar Cell Staining buffer (Fluidigm) followed by 0.8 ml CyFACS. Barcoded samples were combined and pelleted, and then blocked on ice for 15 min with sera from mouse (Thermo Fisher), rat (Thermo Fisher), and human (AB serum, Sigma-Aldrich). Cells were then washed twice with CyFACS, and stained on ice for 45 min with a cocktail of CyTOF surface-staining antibodies (*Supplementary file 1*) in a final volume of 100 µl/well. Cells were then washed three times with CyFACS buffer, and stained on ice for 45 min with the cocktail of lanthanide-conjugated lectins (*Supplementary file 1*) in a final volume of 100 µl/well. Cells were then washed

three times with CyFACS buffer and fixed overnight at 4°C with 2% PFA in metal contaminant-free PBS. The next day, cells were incubated at 4°C for 30 min with fix/perm buffer (eBioscience), and then washed twice with Permeabilization Buffer (eBioscience). After another round of Fc blocking on ice for 15 min with sera from mouse (Thermo Fisher) and rat (Thermo Fisher), cells were washed twice with Permeabilization Buffer (eBioscience), and stained on ice for 45 min with a cocktail of CyTOF intracellular-staining antibodies (*Supplementary file 1*) in a final volume of 100 µl/well. Cells were then washed with CyFACS and incubated for 20 min at room temperature with 250 nM Cell-ID DNA Intercalator-Ir (Fluidigm) in PBS containing 2% PFA. After two more washes with CyFACS, cells were washed once with Maxpar Cell Staining Buffer (Fluidigm), once with Maxpar PBS (Fluidigm), and once with Maxpar Cell Acquisition Solution (Fluidigm). Immediately prior to sample loading, cells were resuspended to a concentration of $7 \times 10^5$/ml in EQ calibration beads (Fluidigm) diluted 1:9 in Maxpar Cell Acquisition Solution. Cells were acquired on a Helios-upgraded CyTOF2 instrument (Fluidigm) at a rate of 250–350 events/s, at the UCSF Flow Core Facility.

## CyTOF data analysis

Data were normalized to EQ calibration beads and then exported as FCS files. The data were then de-barcoded with CyTOF software (Fluidigm) and imported into FlowJo (BD) for gating. This study's raw datasets, pre-gated on live, singlet events, are available for download via the following link in the Dryad public repository: https://doi.org/10.7272/Q6FT8J92.

Total T cells were identified by sequential gating on intact, live, singlet CD3+ CD19- cells (*Figure 3—figure supplement 1*). Total T cells were then re-exported as FCS files and imported into Cytobank for calculations of MSI, and high-dimensional analyses by t-SNE and FlowSOM. t-SNE and FlowSOM plots were generated with default settings except for a modification of total metaclusters from 10 to 20 for FlowSOM analysis. t-SNE and FlowSOM plots were generated excluding all parameters used upstream in the gating strategy (CD19 and HSA) and all glycan characterization parameters. To map defined populations onto t-SNE plots, subsets were defined by manual gating, and then pseudo-colored on the t-SNE plots using FlowJo software. Box plot graphs were generated using ggplot2 in R.

Identification of PRE cells by PP-SLIDE was implemented using recently described methods (*Ma et al., 2020*; *Neidleman et al., 2020b*) to match each infected cell against every CD4+ T cell in the uninfected sample and using k-nearest neighbor calculations to identify the phenotypically most similar. The degree of enrichment of each FlowSOM cluster in PRE cells was calculated by dividing each cluster's relative size within the PRE cells by its relative size within total uninfected CD4+ Tm cells:

Enrichment ratio (Cluster X) = Number of Cluster X cells relative to PRE cells/Number of Cluster X cells relative to uninfected Tm cells.

Clusters with ratios > 1 were designated as enriched and those with ratios >0 and <1 as non-enriched, while clusters with undetectable PRE cells were not shown.

## Statistical analysis

Expression levels were reported as MSI for each parameter (protein or glycan) within each cell population analyzed. Student's two-sided paired t-tests were used to test for differences in MSI among phenotypic subsets (B cell, CD8+ Tm, CD8+ Tn, CD4+ Tm, and CD4+ Tn cells): among uninfected and bystander cells; or among uninfected cells, PRE cells, and infected cells. p-Values were adjusted for multiple testing using false discovery rate (FDR) via the Benjamini-Hochberg or Holm method as indicated in figure legends. FDR adjusted p-values that were <0.05 were considered as significant.

SLIDE analysis was conducted using the R package SLIDE (*Mukherjee et al., 2018*) as recently described (*Ma et al., 2020*). SLIDE was developed as a nearest-neighbor approach to identify and quantify viral-induced remodeling (*Sen et al., 2014*). The ratios between two distance measures in SLIDE (the remodeling score) provides a relative measure of remodeling, and is compared to a background remodeling score generated from SLIDE analysis of non-infected cells, as recently described (*Mukherjee et al., 2018*).

## Acknowledgements

This work was supported by the National Institutes of Health R01AI127219, R01AI147777, and P01AI131374, UM1 AI164559, and UM1 AI164567 to NRR, and R01DK123733, R01AG062383, R01NS117458, and R21AI143385 to MAM. We also acknowledge NIH for the sorter (S10-RR028962), support from CFAR (P30AI027763), and the James B Pendleton Charitable Trust. We acknowledge the PFCC (RRID:SCR_018206) for assistance in CyTOF data acquisition, enabled by an instrument that was supported in part by the DRC Center Grant NIH P30 DK063720 and NIH S10 1S10OD018040-01. The funders had no role in study design, data collection and analysis, decision to publish, or preparation of the manuscript. We thank Trimble Spitzer for the endometrial specimens; Nicole Lazarus and Eugene Butcher for the Act1 antibody; Claudia Bispo and Stanley Tamaki for CyTOF assistance at the Parnassus Flow Core; and Jane Srivastava and Nandhini Raman for flow cytometry assistance at the Gladstone Flow Core. We also thank Françoise Chanut for editorial assistance, and Robin Givens for administrative assistance.

## Additional information

### Funding

| Funder | Grant reference number | Author |
|---|---|---|
| National Institutes of Health | R01AI127219 | Nadia R Roan |
| National Institutes of Health | R01AI147777 | Nadia R Roan |
| National Institutes of Health | P01AI131374 | Nadia R Roan |
| National Institutes of Health | R01DK123733 | Mohamed Abdel-Mohsen |
| National Institutes of Health | R01AG062383 | Mohamed Abdel-Mohsen |
| National Institutes of Health | R01NS117458 | Mohamed Abdel-Mohsen |
| National Institutes of Health | R21AI143385 | Mohamed Abdel-Mohsen |
| National Institutes of Health | UM1AI164559 | Nadia R Roan |
| National Institutes of Health | UM1AI164567 | Nadia R Roan |

The funders had no role in study design, data collection and interpretation, or the decision to submit the work for publication.

### Author contributions

Tongcui Ma, Conceptualization, Data curation, Formal analysis, Investigation, Methodology, Validation, Visualization, Writing - original draft; Matthew McGregor, Methodology; Leila Giron, Guorui Xie, Ashley F George, Investigation; Mohamed Abdel-Mohsen, Conceptualization, Funding acquisition, Methodology, Resources, Supervision, Writing – review and editing; Nadia R Roan, Conceptualization, Data curation, Funding acquisition, Methodology, Project administration, Resources, Supervision, Validation, Writing - original draft

### Author ORCIDs

Mohamed Abdel-Mohsen (iD) http://orcid.org/0000-0002-9945-4314
Nadia R Roan (iD) http://orcid.org/0000-0002-5464-1976

### Decision letter and Author response

Decision letter https://doi.org/10.7554/eLife.78870.sa1
Author response https://doi.org/10.7554/eLife.78870.sa2

## Additional files

### Supplementary files
• MDAR checklist

• Supplementary file 1. Supplementary tables.

### Data availability
Raw CyTOF data have been deposited in Dryad (https://doi.org/10.7272/dryad.Q6FT8J92).

The following dataset was generated:

| Author(s) | Year | Dataset title | Dataset URL | Database and Identifier |
|-----------|------|---------------|-------------|-------------------------|
| Roan NR | 2022 | Single-cell Glycomics Analysis by CyTOF-Lec Reveals Glycan Features Defining Cells Differentially Susceptible to HIV | https://doi.org/10.7272/Q6FT8J92 | Dryad Digital Repository, 10.7272/Q6FT8J92 |

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
