## [Editor Report]

This study applies a new novel method of single cell detection to biologically relevant systems to try to understand whether glycans on the surface of CD4^+^ T cells impact HIV susceptibility. They find that cells expressing higher levels of fucose and sialic acid are more likely to be infected with HIV than those with low levels. The findings point to glycans as biomarkers and potential determinants for cellular susceptibility to HIV, and open the door to new avenues for studying the interplay between cell surface glycans and viral infections.

---

## [Decision Letter]

**Decision letter after peer review:**

Thank you for submitting your article "CyTOF-Lec: Single-cell glycomics analysis reveals glycan features defining cells differentially susceptible to HIV" for consideration by *eLife*. Your article has been reviewed by 2 peer reviewers, and the evaluation has been overseen by a Reviewing Editor and Satyajit Rath as the Senior Editor. The following individual involved in review of your submission has agreed to reveal their identity: Julie M Overbaugh (Reviewer #1).

Essential revisions:

These suggested revisions are mostly meant for clarity.

Ma et al. take a novel approach to an important problem of host cell susceptibility to HIV. They tackle an understudied area of glycan effects on HIV infection using a new method they developed called CyTOF-lec. This method allows single cell detection of infected cells when using a reporter virus for infection. Importantly, the authors go to considerable trouble to use biologically relevant systems, including a transmitted virus and tonsil, endometrial and peripheral T cells. They find that cells expressing higher levels of fucose and sialic acid are more likely to be infected with to HIV than those with low levels. The studies presented here suggest, although didn't fully resolve, that sialic acid itself may be important for infection in CD4, CCR5 positive cells, although they can't really rule out that sialic acid is simply a biomarker for other cell features, such as activation state and entry receptor levels, which are known to impact susceptibility to HIV. Nonetheless, the findings point to glycans as a biomarker and potential determinant for HIV cells susceptibility and open the door to new avenues for studies the interplay between cell surface glycans and viral infections.

Histograms and dot plots in Figure 1 and S1 show broad unimodal distributions of lectin staining. How much overall information is being gained from these lectin stains? As part of the validation and general description of staining, Figure S1 would benefit from tSNE plots of total CD45 tonsil cells colored by intensities of each of the lectins (perhaps annotated by major subsets). This would be useful to provide a sense of how much the staining with each of these lectins varies and co-varies across broad cell subsets. Along similar lines, some representative histograms for lectin staining would be good to add for key figures showing differences in lectin staining (e.g., differences plotted in Figure 3). Also, for Figure 4, please show how high vs. low staining cells were gated.

It's confusing that both signal intensity (MSI) and gating are used to analyze data in Figures3-5

Did the authors perform replicates and quantify changes caused by Sialidase treatment? Please indicate the number of replicates

It is surprising how variable the scales are across tissues and this raises concerns about the possibility of technical reason for this. For instance, that tonsil cells have 10-old lower AOL binding than PBMCs (in several figures including Figure 4C). Were these tissue samples stained and run in parallel in order to test for these differences? Could there be difference in the way that the cells were collected or processed that might be behind this? Conclusions from the section ending on line 262 are not clear.

The tSNE plots provided throughout are poorly annotated in terms general features of the different regions of the plot. Where possible, some effort to better annotate major populations within these plots would be appreciated. It would also be appreciated if the authors limited the number of unique tSNE embedding used. For instance, was tSNE run in parallel the plots shown in Figure 5? If not, if possible it would be great if different tissues could be merged unless batch effects a major issue for these analyses? A general trend in high dimensional analysis has been towards the use of UMAP, which are often more intuitive and easier to annotate. Did the authors consider using UMAP instead of tSNE?

tSNE plots colored by AOL and WGA staining in Figure 4C, 5C are difficult to read. Larger dot size and font sizes for scale labels would be helpful.

The weakest data in Figure 6 J, where increasing removal of sialic acid has just a 2-fold effect on the number of infected cells. Some information in the results of repeat experiments would be helpful as would perhaps studies with higher MOI or more sailadase. If not, the limitations of these findings should be noted.

Related to the point above, why is the effect of sialidase in Figure 1 so modest? A quick explanation there may help explain the limits of the Figure 6 experiment.

For Figure 6, while maybe not inconsistent with the premise of the paper, is it possible that WGA staining is working as a partial proxy for memory phenotype? Tm cells were used as a control but no significant difference between Tm and WGAhigh cells in terms of infection rate is reported in this experiment. While maybe not necessary, the authors could have sorted WGAhigh vs. low cells from within Tm cells to more specifically control for memory status. Similarly, perhaps the analysis in Figure 4D-H could be performed just on non-naïve cells to see if lectin staining differentiates subpopulations of non-naïve cells?

It is good to see the use of corrections for multiple comparisons. They make the findings more robust and impressive.

---

## [Author Response]

Essential revisions:These suggested revisions are mostly meant for clarity.Ma et al. take a novel approach to an important problem of host cell susceptibility to HIV. They tackle an understudied area of glycan effects on HIV infection using a new method they developed called CyTOF-lec. This method allows single cell detection of infected cells when using a reporter virus for infection. Importantly, the authors go to considerable trouble to use biologically relevant systems, including a transmitted virus and tonsil, endometrial and peripheral T cells. They find that cells expressing higher levels of fucose and sialic acid are more likely to be infected with to HIV than those with low levels. The studies presented here suggest, although didn't fully resolve, that sialic acid itself may be important for infection in CD4, CCR5 positive cells, although they can't really rule out that sialic acid is simply a biomarker for other cell features, such as activation state and entry receptor levels, which are known to impact susceptibility to HIV. Nonetheless, the findings point to glycans as a biomarker and potential determinant for HIV cells susceptibility and open the door to new avenues for studies the interplay between cell surface glycans and viral infections.

We thank the editor and reviewers for the nice summary, and for the positive response to our study.

Histograms and dot plots in Figure 1 and S1 show broad unimodal distributions of lectin staining. How much overall information is being gained from these lectin stains?

Indeed, all the lectins had stained with a unimodal distribution pattern. In fact, many protein phenotyping markers (e.g., the CCR7 and CD27 antigens commonly used to define the Tcm and Tem subsets) also stain unimodally, in contrast to the distinct bimodal staining one gets with CD4 or CD8. Importantly, there is a lot of valuable information to be gained from unimodal staining patterns (including by the lectins), since in high-dimensional space it becomes clear that such staining is not random but very specific for different subsets or clusters of cells. This is now visualized in the new tSNE plots we added as part of Figure S1 (now re-named Figure 1—figure supplement 1 per *eLife*’s format) per the request immediately below.

As part of the validation and general description of staining, Figure S1 would benefit from tSNE plots of total CD45 tonsil cells colored by intensities of each of the lectins (perhaps annotated by major subsets). This would be useful to provide a sense of how much the staining with each of these lectins varies and co-varies across broad cell subsets.

As requested, we have now added to Figure S1A (now re-named Figure 1—figure supplement 1 per *ELife*’s format) tSNE plots of the immune cells colored by intensities of each of the lectins annotated by the major subsets.

Along similar lines, some representative histograms for lectin staining would be good to add for key figures showing differences in lectin staining (e.g., differences plotted in Figure 3).

We have now shown representative histograms for lectin staining for the key differences we reported, and added this as the new Figure 3—figure supplement 4. As histograms provide less information than depiction of lectin staining via tSNE heatmaps (where relative lectin staining on individual cells can be visualized), we additionally show in this new supplementary figure the same data as tSNE plots. Both forms of visualization support the original conclusions drawn in Figure 3.

Also, for Figure 4, please show how high vs. low staining cells were gated.

This has now been added as a new panel in Figure 4 (Figure 4A).

It's confusing that both signal intensity (MSI) and gating are used to analyze data in Figures3-5

We typically perform both MSI and manual gating for all our analyses, since we find that when these two ways of analyzing the same dataset lead to the same conclusions, it makes us more confident in drawing those conclusions. In Figures 3 and 4, MSI and manual gating for cells expressing different levels of glycans led to the same conclusions. Of note, Figure 5 did not entail MSI nor manual gating analysis, but rather clustering.

Did the authors perform replicates and quantify changes caused by Sialidase treatment? Please indicate the number of replicates

For each of the 2 donors analyzed, experimental triplicates were implemented in the infection assays, as stated in the figure legend of Figure 6K. For quantifying changes in sialic acid levels induced by sialidase treatment, we had previously shown the data for only one of the two donors (Figure 6I). We now show in the new Figure 6I the effect of sialidase treatment on WGA binding for both donors, where in both instances there is a clear dose-dependent decrease in WGA binding upon sialidase treatment.

It is surprising how variable the scales are across tissues and this raises concerns about the possibility of technical reason for this. For instance, that tonsil cells have 10-old lower AOL binding than PBMCs (in several figures including Figure 4C). Were these tissue samples stained and run in parallel in order to test for these differences? Could there be difference in the way that the cells were collected or processed that might be behind this?

The tonsil specimens were run in different batches from the endometrial and blood specimens, so we cannot formally exclude that the lower staining is due to batch effects. That being said, given the consistency of the lower staining by some (but not all) of the lectins, and not other CyTOF protein-based markers, we believe that tonsils may indeed express lower levels of some of the glycans (e.g., that recognized by AOL). It was unfortunately not logistically possible to run the tonsil specimens within the same batch as the endometrial and PBMC specimens. We have now acknowledged (Lines 430-433) that a limitation of our study was the need to run specimens in multiple batches, which resulted in the tonsil, endometrial, vs. PBMC specimens to be run in different batches.

Conclusions from the section ending on line 262 are not clear.

We have now added a sentence to sum up the overall conclusion from this section: “All together, these results suggest that among blood and tonsillar memory CD4^+^ T cells, those with the highest levels of fucose and sialic acid are preferentially targeted for infection by HIV; this phenomenon was however not observed among endometrial cells.” (Lines 269-271).

The tSNE plots provided throughout are poorly annotated in terms general features of the different regions of the plot. Where possible, some effort to better annotate major populations within these plots would be appreciated.

The new tSNE plots showing lectin staining on total immune cells (Figure S1, now renamed Figure 1—figure supplement 1 per *ELife*’s format), as requested above, now clearly depict the major immune subsets. Other tSNE plots throughout the manuscript were pre-gated on memory CD4^+^ T cells, and therefore are difficult to annotate further as they are already a relatively small subset. Sub-subsets of memory CD4^+^ T cells typically do not segregate as distinct “islands” on tSNE plots and therefore is not easily to annotate in a visual manner.

It would also be appreciated if the authors limited the number of unique tSNE embedding used. For instance, was tSNE run in parallel the plots shown in Figure 5? If not, if possible it would be great if different tissues could be merged unless batch effects a major issue for these analyses?

We in fact initially tried using the same tSNE “space” for visualization of data generated from the 3 compartments (blood, tonsil, endometrium), but visualization that segregated the cells by each of the compartments, as demonstrated in Author response image 1:

**Author response image 1. sa2fig1:** 

In Author response image 1, we are depicting the same datasets shown in Figure 5, but run within the same tSNE space, and color-coded according to expression levels of the memory marker CD45RO. This way of visualizing the data is not effective as each compartment has less “space” for visualization, since much space is taken up by cells from a different compartment. This is particularly true for the endometrial cells, which segregated into a very small area of the tSNE when analyzed in this manner. By contrast, when run as its own tSNE, as we had shown in Figure 5, the endometrial cells nicely separate out into distinct visualizable clusters.It is for this reason that we ran separate tSNE plots for each compartment.

A general trend in high dimensional analysis has been towards the use of UMAP, which are often more intuitive and easier to annotate. Did the authors consider using UMAP instead of tSNE?

We had considered use of UMAP, and indeed in other unrelated work analyzing single-cell RNAseq datasets we routinely use UMAP for data visualization. UMAP, however, like tSNE, has its disadvantages, including that it does not preserve local neighborhoods. Because CyTOF datasets are still commonly depicted by tSNE, and because CyTOF analysis software and scripts commonly use tSNE visualization, we decided to stick with tSNE visualizations. This also enables better visual comparisons with our prior CyTOF datasets on HIV specimens which all implemented tSNE visualizations (e.g., PMID 35296537, PMID 35154118, PMID 33910003, PMID 32990219, PMID 32452381, PMID 28746881).

tSNE plots colored by AOL and WGA staining in Figure 4C, 5C are difficult to read. Larger dot size and font sizes for scale labels would be helpful.

We have made larger dot and font sizes, as requested.

The weakest data in Figure 6 J, where increasing removal of sialic acid has just a 2-fold effect on the number of infected cells. Some information in the results of repeat experiments would be helpful as would perhaps studies with higher MOI or more sailadase. If not, the limitations of these findings should be noted.

These sialidase treatment + infection experiments were performed on two donors. These experiments were technically challenging as sialidase treatment proved to be rather toxic for primary CD4^+^ T cells (even after a 1 hour treatment). We have now added a discussion point acknowledging the limitations of these findings as they were generated with only two donors (Lines 438-441).

Related to the point above, why is the effect of sialidase in Figure 1 so modest? A quick explanation there may help explain the limits of the Figure 6 experiment.

The effects of sialidase were modest likely because treatment was for only one hour. As mentioned above, sialidase treatment of primary cells proved to be rather cytotoxic, and even after one hour of treatment there was already significant toxicity observed in some of the tested donors. As mentioned above, this limitation has now been acknowledged in the discussion.

For Figure 6, while maybe not inconsistent with the premise of the paper, is it possible that WGA staining is working as a partial proxy for memory phenotype? Tm cells were used as a control but no significant difference between Tm and WGAhigh cells in terms of infection rate is reported in this experiment.

We apologize for the confusion, but in fact we had pre-purified for memory CD4^+^ T cells (CD4^+^ Tm) prior to sorting for the WGAhigh, WGAmed, and WGAlow cells. This was indicated by the pre-purification step as described in the methods section, which we have now further emphasized by re-iterating in that section the notion of starting with purified memory CD4^+^ T cells (Line 545). Our figure legend for this experiment (Figure 6A) had also indicated that we had started with CD4^+^ Tm cells, and we have now added in additional indications of this in the legend for Figure 6B-C to avoid any ambiguity (Lines 730-732).

We had set up the experiment this way to ensure that any effect we observed was not due to WGA simply enriching for memory cells, since in agreement with the Reviewer this would not be very meaningful. Our observation that WGAlow cells within the memory CD4^+^ T cell subpopulation have low infection rates relative to total memory CD4^+^ T cells demonstrate that these *memory* CD4^+^ T cells binding low levels of WGA are less permissive to infection than total memory CD4^+^ T cells.

While maybe not necessary, the authors could have sorted WGAhigh vs. low cells from within Tm cells to more specifically control for memory status.

Please see above, this was in fact the way the experiment was originally performed (sorting from within Tm cells WGAhigh, WGAmed, and WGAlow).

Similarly, perhaps the analysis in Figure 4D-H could be performed just on non-naïve cells to see if lectin staining differentiates subpopulations of non-naïve cells?

We assume the Reviewer is referring to Figure 6D-H (since there was no Figure 4E, F, G, and H in the original submission). In fact, Figure 6D-H were performed on just non-naïve CD4^+^ T cells (in other words, memory CD4^+^ T cells). In the figure legend of these panels, we always referred to “Tm cells” which is our abbreviation for “memory” as defined earlier in the manuscript.

It is good to see the use of corrections for multiple comparisons. They make the findings more robust and impressive.

We thank the Editor/Reviewers for recognizing this!